# Understanding variability in root zone storage capacity in boreal regions

Tanja de Boer-Euser[1,2], Leo-Juhani Meriö[3], and Hannu Marttila[3]

[1]Water Resources Section, Faculty of Civil Engineering and Geosciences, Delft University of Technology, P.O. Box 5048, NL-2600 GA Delft, The Netherlands
[2]Department of Civil Engineering, Eduardo Mondlane University, C.P. 257 Maputo, Mozambique
[3]Water Resources and Environmental Engineering Research Unit, PO Box 4300, 90014 University of Oulu, Finland

**Correspondence:** Tanja de Boer-Euser (tanjaeuser@gmail.com)

**Abstract.** The root zone storage capacity ($S_r$) of vegetation is an important parameter in the hydrological behaviour of a catchment. Traditionally, $S_r$ is derived from soil and vegetation data. However, more recently a new method has been developed that uses climate data to estimate $S_r$ based on the assumption that vegetation adapts its root zone storage capacity to overcome dry periods. This method also enables one to take into account temporal variability of derived $S_r$-values resulting

from changes in climate or land cover. The current study applies this new method in 64 catchments in Finland to investigate the reasons for variability in $S_r$ in boreal regions. Relations were assessed between climate derived $S_r$-values and climate variables (precipitation-potential evaporation rate, mean annual temperature, max snow water equivalent, snow-off date), detailed vegetation characteristics (leaf cover, tree length, root biomass), and vegetation types. The results show that especially the phase difference between snow-off date and onset of potential evaporation has a large influence on the derived $S_r$-values. Further to

this it is found that (non-)coincidence of snow melt and potential evaporation could cause a division between catchments with a high and a low $S_r$-value. It is concluded that the climate derived root zone storage capacity leads to plausible $S_r$-values in boreal areas and that apart from climate variables, catchment vegetation characteristics can also be directly linked to the derived $S_r$-values. As the climate derived $S_r$ enables incorporating climatic and vegetation conditions in a hydrological parameter, it could be beneficial to assess the effects of changing climate and environmental conditions in boreal regions.

## 1 Introduction

The hydrological cycle of boreal regions is changing vastly as a result of climate change (Prowse et al., 2015) and increasing anthropogenic land use activities (Instanes et al., 2016). Increasing temperatures and precipitation, shifts in precipitation from snow to rainfall and retreating seasonal snow cover are a few examples of alterations of the boreal hydrological cycle (Bring et al., 2016). Consequences of increasing temperatures are likely to be most severe in boreal systems, as slight changes in

temperature can alter the magnitude and timing of snow accumulation and melt (Carey et al., 2010). Predicted changes create climatic conditions at certain higher latitudes, which are similar to those at lower latitudes a few decades earlier (Intergovernmental Panel on Climate Change, 2014). These changes in climate will have an effect on different vegetation types while at the same time, land use activities have been intensified especially in European countries and are predicted to increase in

the near future due to a "green shift" to a bio-based economy (Golembiewski et al., 2015). The land use changes consist of modifications in actual land use (increase in forest cover), but also of more intensive use of forests, including clear cutting, forest trimming, residual harvest and of increasing utilisation of peatland forests as a source for biomass (e.g. Laudon et al., 2011; Nieminen et al., 2017).

Especially under these changing conditions, a proper hydrological understanding of boreal catchments is needed (Waddington et al., 2015; Laudon et al., 2017) to understand the sensitivity and resilience of catchments (Tetzlaff et al., 2013), but also to assess the effect of possible land use activities. Many studies have been conducted to explore hydrological changes resulting from land use activities (Ide et al., 2013; Mannerkoski et al., 2005; Nieminen et al., 2017), and some already studied changes in transpiration (patterns) at the catchment scale in boreal regions (e.g. van der Velde et al., 2013; Jaramillo et al., 2018). The

partitioning between transpiration and runoff is largely determined by the water use efficiency of vegetation (e.g. Troch et al., 2009) and the available root zone storage capacity ($S_r$) of the vegetation (e.g., Zhang et al., 2001): the water use efficiency determines the amount of water the vegetation needs and the root zone storage capacity ensures sufficient storage to supply this water. Thus, detailed knowledge about these variables can increase the hydrological understanding of catchments under different conditions.

Traditionally, $S_r$ is estimated from soil and vegetation data or calibrated in a hydrological model. Following the analysis that $S_r$ is strongly related to climate variables (e.g., Kleidon and Heimann, 1998; Gentine et al., 2012; Gimbel et al., 2016), Gao et al. (2014) developed a new method to estimate $S_r$ from climate data. Subsequently, several studies have been carried out in which this method was used. For example, Wang-Erlandsson et al. (2016) used earth observation data to estimate $S_r$ globally; de Boer-Euser et al. (2016) did a comparison between the influence of soil and climate on $S_r$; Nijzink et al. (2016)

investigated the change in $S_r$ after deforestation and Zhao et al. (2016) introduced a snow component to the method and carried out a sensitivity analysis.

Thus, climate (or the balance between precipitation and transpiration) has a large influence on the developed $S_r$. However, it is very likely that root development is affected by other factors, including nutrients (e.g., Shahzad and Amtmann, 2017), the survival mechanism of the vegetation (e.g., Christina et al., 2017), or reduced space for root development due to shallow soil

layers or high ground water tables (e.g., Soylu et al., 2014). $S_r$ is expected to change if any of these factors changes, which has consequences for the hydrology of the area (e.g., Saft et al., 2015). Assessing the (future) hydrology of boreal catchments could benefit from a better understanding of the relation between $S_r$ and climatic and vegetation conditions.

The method to derive $S_r$ from climate data was originally developed to estimate an important parameter in conceptual hydrological models (e.g. Gao et al., 2014). So, influences on the derivation and wider applicability of the climate derived

$S_r$ need to be investigated before it can be used to further assess the hydrology of boreal areas and to assist in assessing the hydrological effects of climatic and land use changes. Therefore, this study aims at better understanding the influences of different climate variables on the climate derived $S_r$-values and its wider applicability by comparing it with various catchment and vegetation characteristics.

## 2 Methods

### 2.1 Characteristics of study catchments

A total of 64 headwater catchments were used for this study, spread over Finland. The catchments are located in different boreal regions (south, mid- and north boreal; Ahti et al., 1968) and thus have different climate conditions and vegetation patterns (Figure 1). All catchments belong to a national network of small catchments (Seuna and Linjama, 2004) and have been used in various studies (e.g., Kortelainen et al., 2006; Sarkkola et al., 2012, 2013b). The catchments used in this study were selected based on the availability of long-term runoff records, snow line records and meteorological data from the catchments.

The climate of the region is humid, with annual average air temperatures varying from 5 °C in the south to -2 °C in the north and average precipitation of 600-700 mm/y in the south and 450-550 mm/y in the north. Average maximum snow depth by the end of March is 50-400 mm in the south and 600-800 mm in the north.

The principal land cover in the study catchments is forest (with a median of 81% coverage of evergreen, deciduous and mixed forest), followed by shrubs and herbaceous vegetation, inland waters and wetlands. Agricultural activities were present in some of the catchments in the south and mid-boreal regions. Total root biomass, as well as root biomass for spruce and deciduous trees decreases towards the north, while pine root biomass is more or less constant (Figure 1). The surface area of the catchments ranges from 0.07 km$^2$ to 122 km$^2$ (median 6.15 km$^2$).

The soil type in the southern catchments is dominated by clay layers whereas basal till and peatland cover is increasing when moving towards east and north. The catchments have relatively flat topography with a mean difference in elevation of approximately 70 m. The selected catchments do not contain any urban settlements. Tables 1 and 2 in the supplementary material give an overview of available vegetation and climate characteristics for the study catchments.

### 2.2 Data use and correction

Two sets of data were used in the study: one for the calculation of the climate derived root zone storage capacity and one to investigate the variation of $S_r$. For the $S_r$ calculations daily precipitation, daily snow water equivalent, monthly potential evaporation and yearly discharge data were used. For investigating the variability and relations with catchment characteristics additional data were used, including leaf cover, tree length, root biomass, temperature, snow-off date and vegetation type.

Daily discharge was measured with water stage recorders and weirs were routinely checked for errors by the Finnish Environment Institute. Precipitation ($P$) and temperature data were taken from the national 10 km x 10 km interpolated grid produced by the Finnish Meteorological Institute (FMI) (Paituli database[1]). These data have been checked for measurement errors caused by gauges and were corrected in operative quality control. The snow line data for snow water equivalent ($S_{SWE}$), potential evaporation ($E_p$), using pan measurements, and runoff data used were obtained from the Finnish Environmental Institute's open database (Hertta). Note that because $E_p$ is derived from pan measurements, it is not measured when temperatures are below zero. However, it can be assumed that if it would be measured, amounts would be very low.

---

[1] https://avaa.tdata.fi/web/paituli/latauspalvelu

The snow line measurement points were either located inside or in close proximity of the study catchments; however, for some catchments the increase in $S_{SWE}$ during a season was higher than the total measured precipitation for the same period. As the precipitation data were assumed to be more reliable and less spatially variable, the $S_{SWE}$ data were adjusted on a daily basis to make them consistent with the precipitation data.

Corine Land Cover 2012 data (Paituli database) was used for determining the vegetation types occuring in the study catchments. The surface lithology and geology data are based on the Surface Geology Map of Finland (Hakku database[2]). Data for root biomass, tree length and leaf cover are based on multi-source national forest inventory data provided by the Natural Resources Institute Finland (LUKE open data[3]). Data are based on field inventory data, satellite images, digital map data and other georeferenced data sets (for more information refer to Mäkisara et al., 2016). Tree data were available for Pine, Spruce

and Deciduous forest types. Drained and pristine peatlands masks were obtained from the Finnish Environmental Institute (SYKE).

## 2.3    Climate derived root zone storage capacity

To investigate the variability in root zone storage capacity, a climate derived root zone storage capacity ($S_r$) was used. The derivation of this $S_r$ is based on the principle that vegetation will create a buffer with its root system just sufficient to overcome

a drought with a certain return period. Investing less in a root system would lead to the vegetation dying in case of a severer drought and investing more is not efficient in terms of carbon use. This method results in a catchment representative storage capacity, which reflects the root zone storage capacity for all vegetation combined in a catchment. It is further assumed that the amount of required storage depends on the amount of water that should have transpired to close the water balance. In this study the same base calculation was used as in de Boer-Euser et al. (2016), but as snow accumulation cannot be neglected

in Finland, an additional snow module was added (Figure 2). For the calculation of $S_r$ the daily balance between infiltration ($I$) and transpiration demand ($T$) is used to simulate the amount of storage the vegetation would need to cover the infiltration deficit.

    The transpiration demand used in this method is the amount of water that should, in the long term, transpire to close the water balance. For the calculation $\overline{T}$ was thus derived from the long term water balance ($\overline{T} = \overline{P} - \overline{E_i} - \overline{Q}$); following monthly

averaged potential evaporation was used to add seasonality to $T$. Infiltration was assumed to be the result of precipitation minus interception evaporation in the original calculations (e.g., Gao et al., 2014; de Boer-Euser et al., 2016). However, in case of solid precipitation, the precipitation is stored on the soil surface for days to months and only infiltrates during the snow melt period. As this is a relevant process in most of the study catchments, a snow component (Equations 1-4) was added to the calculation method. The change in $S_{SWE}$ was used to determine the amount of precipitation stored on and infiltrating into the

soil on a daily basis. Interception was only taken into account in case of liquid precipitation and an interception threshold of 1.5 mm was assumed for all catchments. Sublimation was not taken into account, as potential evaporation is generally (very) low when snow cover is present.

---

[2]https://hakku.gtk.fi/en/locations/search

[3]http://kartta.metla.fi/opendata/valinta.html

The estimates for infiltration and transpiration demand were used in a daily simulation of the root zone storage. Infiltration forms the inflow of water and transpiration the extraction; any excess water is assumed to runoff directly. This simulation results in annual required maximum storage capacities, which were used in a Gumbel distribution (Gumbel, 1935) to obtain the required storage capacity to overcome a drougth with 20-year return period. A 20-year return period was selected as an averaged catchment representative, following the results of Gao et al. (2014) and Wang-Erlandsson et al. (2016) and based on the high percentage of forest cover in the study catchments.

The method described above estimates $S_r$ for a current situation based on historical drought occurrences. However, the same principle and calculation method can be used to estimate $S_r$ under changing conditions. These can be derived from observed data (e.g. Nijzink et al., 2016), but can also consist of scenarios of changing climate variables or land use characteristics. The latter one could be represented by using a different drought return period (e.g. Wang-Erlandsson et al., 2016).

For estimating $S_r$ in this study, data from 1 January 1990 to 31 December 2012 were used. For precipitation and snow water equivalent daily values were used, while for discharge and potential evaporation data, long term yearly and monthly average were used respectively. For some of the catchments discharge data were limitedly available for the study period; for these catchments older discharge data were taken into account as well to obtain a long term average.

$$P_{rz} = P_i + P_m \tag{1}$$

$$P_i = \begin{cases} 0, & \text{if } S_{SWE} > 0 \text{ and } \Delta S_{SWE} < 0 \\ 0, & \text{if } S_{SWE} > 0 \text{ and } \Delta S_{SWE} > 0 \\ P_t, & \text{if } S_{SWE} = 0 \end{cases} \tag{2}$$

$$P_m = \begin{cases} P_t - \Delta S_{SWE}, & \text{if } S_{SWE} > 0 \text{ and } \Delta S_{SWE} < 0 \\ 0, & \text{if } S_{SWE} > 0 \text{ and } \Delta S_{SWE} > 0 \\ 0, & \text{if } S_{SWE} = 0 \end{cases} \tag{3}$$

$$\Delta S_{SWE} = S_{SWE,t=i} - S_{SWE,t=i-1} \tag{4}$$

with, $P_{rz}$ = infiltration, $P_t$ total precipitation, $P_i$ effective precipitation, $P_m$ snow melt, $S_{SWE}$ snow water equivalent.

## 2.4 Relations between $S_r$ and catchment characteristics

To further explore the physical meaning and applicability of the climate derived root zone storage capacity, $S_r$-values were compared with climate variables, vegetation characteristics and coverage of vegetation types.

### 2.4.1 Climate variables

The method used to derive $S_r$ is based on climate data, so it is expected that climate has a strong influence on the derived $S_r$-values. However, the derived $S_r$-values are not a linear combination of the used variables (i.e. daily $P$, daily $S_{SWE}$, yearly $Q$, monthly $E_p$) and thus the influence of different climate variables is not straight forward. Therefore, derived $S_r$-values are compared with four other climate variables ($P/E_p$-ratio, mean annual temperature, snow-off date and maximum $S_{SWE}$) to analyse which ones have the strongest relation with the $S_r$-values. These variables were selected as they are expected to reflect the absolute and phase difference between water supply (precipitation and snow melt) and water demand (transpiration), which is assumed to have the largest influence on the derived $S_r$-values.

First, the relations between the estimated $S_r$-values and climate variables were assessed by analysing spatial patterns and scatterplots. To assess the correlation between the different variables, the non-parametric Spearman's correlation coefficient was used.

### 2.4.2 Vegetation characteristics

The climate derived $S_r$ is originally a parameter for conceptual hydrological models and for that purpose it is expected to reflect a representative storage capacity in a catchment. In that sense it cannot be attributed to a single type of vegetation or be directly measured in the field; despite this, it is expected that it is related to actual vegetation characteristics. When this correlation indeed exists, the climate derived $S_r$ will be more useful to use for other purposes than modelling.

First, it is expected that vegetation actually has to increase its root biomass in order to increase the root zone storage capacity. Therefore, the derived $S_r$ is compared with data about root biomass for three different tree types. Second, an essential part of the $S_r$ calculation is the estimation of the transpiration demand. The average transpiration for the calculations is derived from the water balance (difference between precipitation and discharge), and is reflected in the derived $S_r$-values. As the precipitation is relatively similar for the study catchments (mean of 1.65 mm/d, with a standard deviation of 0.14 mm/d), higher transpiration demands will lead to higher $S_r$-values. Similarly, higher transpiration demands indicate that the vegetation can use more (solar) energy for their development and thus, establishing more above ground biomass as well. So, it is expected that the derived $S_r$-values are related to vegetation properties like leaf cover and tree height as well.

### 2.4.3 Vegetation types

Different vegetation types and their corresponding land covers occur in different climates and ecosystems and can have different survival mechanisms. And, a change of vegetation or land cover type is likely to change the transpiration and thus the hydrology of a catchment. Therefore, the relation between $S_r$ and land cover and vegetation types was investigated. The vegetation types included in this analysis are forest (containing all forest types), pristine peatlands, drained peatlands (covered with either forest or agriculture) and agricultural area. The relations between the estimated $S_r$-values and these vegetation types were assessed using scatterplots between $S_r$ and the vegetation types. The non-parametric Spearman's correlation coefficient was used to assess the correlation between the different variables.

#### 2.4.4 Correlations among catchment characteristics

The catchment characteristics that were compared with the climate derived $S_r$ are very likely to be correlated, making it difficult to assess their individual relation with $S_r$. A principal component analysis (PCA) was set up across all catchments to explore the dependencies between the used characteristics. A PCA is a statistical tool which can be used to reduce the dimensions of a problem and explore correlations between variables.

Before carrying out the PCA, the end products were standardised to have zero mean and unit variance on the covariance matrix. The final number of principal components (PCs) was determined using the broken-stick model (Jackson, 1993), in which eigenvalues from a PCA are compared with the broken-stick distribution. Since each eigenvalue of a PCA represents a measure of a component's variance, a component was retained if its eigenvalue was larger than the value given by the broken-stick model. Numerical results of the PCA can be found in Table 3 in the supplementary material.

## 3 Results

### 3.1 Climate variables

Derived root zone storage capacities were compared with a set of climate variables reflecting the absolute and phase difference between water supply and demand. Focusing first on the relation between $S_r$ and the absolute difference, Figure 3 shows the spatial patterns of $S_r$ and $P/E_p$ (a definition of the aridity index). $S_r$-values generally decrease from south to north and especially for the mid-boreal region a large difference exists between the eastern and western side of the country. For the catchments in the north and mid-boreal regions larger $S_r$-values generally coincide with smaller $P/E_p$ ratios, but for the south boreal region this pattern is less clear. The same can be observed from Figure 4a: the catchments in the north and mid-boreal regions show a negative correlation between $S_r$ and $P/E_p$, while in the south boreal region no significant correlation exists: the range in $S_r$-values is large, although the variability in $P/E_p$ is small.

Second, snow cover (expressed in snow water equivalent, $S_{SWE}$) is important when focusing on the phase difference between water supply and demand. With more precipitation being stored for longer periods the supply of water will be delayed. Figure 3 shows for the majority of the catchments higher derived $S_r$-values (a) in case of lower maximum $S_{SWE}$ (b). However, for some catchments in the mid-boreal region very small $S_r$-values are derived while maximum $S_{SWE}$ is not very high. As also discussed in section 3.4 and shown in Figure 8 $P/E_p$ and $S_{SWE}$ are correlated. Especially, both $E_p$ and snow storage and melt are driven by temperature. Figure 4 shows the strongest correlation between mean annual temperature ($T_{MA}$) and $S_r$, followed by snow-off date, maximum $S_{SWE}$ and $P/E_p$. This indicates that for the studied catchments the phase difference as well as the absolute difference between water supply and demand are important, with the first one probably having a larger influence.

## 3.2 Vegetation characteristics

Estimated root zone storage capacities were compared with characteristics of the vegetation in the study catchments. In Figure 5 $S_r$ is compared with the observed root biomass in the catchments. A distinction is made between three types of trees: pine, spruce and deciduous trees. Root biomass of spruce and deciduous trees is positively correlated with $S_r$ when considering all catchments; when considering the individual boreal regions, only a significant correlation exists for deciduous trees in the north boreal region. The correlation between $S_r$ and root biomass of pine is very interesting: a negative correlation exists between $S_r$ and root biomass when considering all catchments. For the individual regions no significant correlation exists. This finding indicates that more storage is created with less or thinner roots. Figure 5d combines the results for all tree types and shows in general higher $S_r$-values for higher densities of root biomass, but this correlation is not significant.

Figure 6 shows the relation between $S_r$ and average leaf cover (top row) and tree height (bottom row). For both comparisons the data is plotted indicating the occurence of different vegetation types (forest, pristine peatlands and agriculture) in the catchments and the boreal regions in which the catchments are located. $S_r$ is positively correlated with both leaf cover and tree height (Spearman's coefficients of 0.33 and 0.32 respectively), but no significant correlation exists for the individual boreal regions. When looking at the different vegetation types, it can be seen that catchments with a large forest cover are the ones with the widest range in leaf cover and tree height. Especially for catchments with a large agricultural cover this range is smaller. More details about the relation between vegetation type and $S_r$ are discussed in Section 3.3 and Figure 7.

## 3.3 Vegetation types

In addition to climate and vegetation characteristics, also vegetation types can have an influence on the derived $S_r$, mainly because different vegetation types have different transpiration patterns and survival strategies. Before analysing correlations between $S_r$ and vegetation type, it should be noted though that vegetation types are (partly) correlated with climate as well (Figure 8). This is especially relevant for the correlations between $S_r$ and (pristine) peatlands and agriculture.

The strongest correlation between $S_r$ and vegetation types can be found for agricultural covers; here not only a significant positive correlation is present when considering all catchments, but also for the three individual regions (Figure 7). Further, a decrease in forested area coincides with a larger range in $S_r$, but no significant correlation is found, neither for all catchments and for the individual regions (Figure 7b). The drained peatlands (Figure 7c) also show a negative correlation with $S_r$ when considering all catchments and for the mid-boreal region: for the north and south boreal regions no significant correlations were found. While for the former three vegetation types a stronger or weaker gradual relation with $S_r$ can visually be observed, the pristine peatlands show strong threshold behaviour. For catchments covered for more than 20% with pristine peatlands, $S_r$-values are below 115 mm. It should be noted though, that catchments with high pristine peatland cover do not occur in the south boreal region.

### 3.4 Correlations among catchment characteristics

The variables that were compared with $S_r$ are very likely to be correlated among themselves as well. Therefore, Figure 8 shows a principal component analysis based on the catchment characteristics used in the analysis. Figure 8a shows the individual catchments with their loadings on PC1 and PC2 (with a combined explained variance of 54%); Figure 8b shows the same for the catchment characteristics used in the comparison. The plotted catchments (a) indicate that the eco-regions mainly differ in climate characteristics and that especially in the mid- and south boreal regions a large range of vegetation characteristics and vegetation types occur.

Figure 8b shows that the majority of the climate variables (shown in blue) are positively correlated to each other and negatively correlated to the mean annual temperature and transpiration demand. What can also be seen is the limited correlation between the majority of the climate variables and (summer) precipitation. With respect to vegetation characteristics (shown in green), these are strongly correlated with forest and agricultural land covers, but limitedly correlated to the majority of the climate variables. Only peatland covers are positively correlated with the majority of the climate variables.

Especially, the relative independence of the vegetation characteristics and vegetation types with respect to the climate variables is important to keep in mind when interpreting the results. This means that relations between $S_r$-values and vegetation characteristics are not likely to be strongly influenced by the climate variables.

### 3.5 Threshold behaviour

The results presented before show to a variable extent a threshold in the relation between the derived $S_r$-values and the catchment characteristics. This threshold is mainly visible in Figures 4 and 7d and seems to be the strongest for snow characteristics (Figure 4c,d) and pristine peatlands (Figure 7d). For all variables the threshold is located at a $S_r$ of approximately 115 mm. To further investigate the origin and position of the threshold the catchments were divided into two groups separated by a $S_r$ of 115 mm. Within the groups statistically significant variations exist in both vegetation, specifically in tree root biomass (pine RBM: Mann-Whitney U-test, p=0.0131; spruce RBM: U-test, p=0.0363) and proportion of pristine (U-test, p=0.0008) and drained (U-test, p=0.0135) peatlands. At the same time also climatic parameters changed: $P/E_p$ (U-test, p=0.0264), max $S_{SWE}$ (U-test, p=0.0000), snow-off date (U-test, p=0.0000) and mean annual temperature ($T_{MA}$: U-test, p=0.0000) showed a significant difference between the groups.

As not only the maximum $S_{SWE}$ and $T_{MA}$ show a strong correlation with $S_r$, but also the snow-off date (Figure 4), it is possible that the threshold is related to the phase difference between water input and demand in the catchments. Therefore, Figure 9 shows the period with snow cover (colour plot) and the period in which potential evaporation is above zero (white lines) for each catchment. In general, for catchments with a $S_r$ smaller than 115 mm (bottom part of the plot), the snow melt and onset of potential evaporation overlap. On the other hand, for catchments with a $S_r$ larger than 115 mm the snow has already melted at the onset of the potential evaporation measurements. In the first case the phase difference between input and demand is decreased, while in the second case it is increased, thus requiring a larger storage capacity. The phase difference between snow-off and onset of $E_p$ was calculated and included in Figure 8; it is positively correlated with the majority of the other

climate variables. It is therefore likely to show the combined effect of the different climatic influences. This phase difference gives an explanation for the origin of the threshold, but not for the location at 115mm. A clear reason for the threshold being located at 115 mm could not be found and it might be an artifact of this specific data set.

## 4  Discussion

The presented results show that among the compared characteristics the climate derived root zone storage capacities are strongest related to climate variables, followed by vegetation characteristics and vegetation types. These results gain better understanding of the influence of the different climate variables on the calculation of $S_r$ in snow dominated regions. The boreal ecosystems has been referred as a "green desert" (e.g., Hall, 1999; Betts et al., 2001); although ample water is available on the surface, the vegetation is less productive and evaporation rates are generally low, because of either nutrient limitations
or adaptation to cool environments. Our results can thus be used to explore the physical meaning and wider application of $S_r$ for land and water management purposes. Below, possible reasons for differences in correlation and for the found threshold are discussed, together with implications of the findings.

### 4.1  Climate variables

As the root zone storage capacity is derived from climate data, logically a correlation exists between the derived $S_r$-values and
various climate variables. The strongest correlations between $S_r$ and the catchment characteristics are found when all three boreal regions are considered together and to a lesser extend when the boreal regions are considered individually; these boreal regions mainly differ in climate characteristics (Figure 8). Together with the results presented in Figure 4 this shows that the relation between climate and $S_r$ is stronger than the relations between $S_r$ and other catchment characteristics.

However, it is interesting to see that not all climate variables have the same influence (Figure 4) on the derived $S_r$-values.
More specifically, the phase difference between the snow-off date (water supply) and onset of potential evaporation (water demand) turns out to be very important (Figure 9). Although the current (non)coincidence of snow-off and the onset of $E_p$ could partly be attributed to the measurement techniques and locations of both variables, it still shows that the derived $S_r$-values are sensitive to the phase difference between the two. Further, the different analyses show that for the colder regions, the influence of individual climate variables ($P/E_p$, $T_{MA}$, snow-off date) is more important. This larger influence of climate
variables in colder regions can also influence or partly cause the observed threshold behaviour.

### 4.2  Vegetation characteristics

Figure 8 shows that the vegetation characteristics are not strongly correlated with the majority of the climate variables, which makes it interesting to compare them with $S_r$. However, the result of this comparison did not show patterns as strong as expected. One of the reasons of this could be the heterogeneity in vegetation types in the study catchments. Another reason
could be that the $S_r$ parameter does not have a very strong physical meaning in boreal regions.

Despite the conceptual character of the climate derived root zone storage capacity, it was expected that it is positively correlated with root density or root biomass; this study is the first to show such a connection exists for spruce and deciduous trees (Figure 5). However, for pine a negative correlation was observed, which means that the vegetation is able to create a larger storage capacity with fewer or thinner roots. This can have multiple reasons, among which, the survival strategies of the trees (e.g., methods to access water or water use efficiency), or the combined effect with other catchment characteristics (e.g., a low density of pine trees in these catchments, thus their influence on the overall transpiration and storage in the catchments or the influence of the drained peatlands in which pine trees often occur). In addition, Figure 5 could also reflect the optimal growing conditions for pine trees: low $S_r$-values coincide with low transpiration demands and thus likely smaller biomass development. On the other hand, for larger $S_r$-values the growing conditions for spruce and deciduous tree become better, thus out competing the pine trees.

By using a climate derived root zone storage capacity, it is assumed that the $S_r$ developed by the vegetation is in balance with the transpiration demands. Not necessarily one causes the other, but a larger $S_r$ coincides with higher or more variable transpiration demands. When the transpiration demands in boreal areas are higher, it is likely that vegetation has higher potential to develop as well (ie. more leaf cover, larger trees). However, if soil conditions are such that root development is slowed down, but still vegetation survives, it is likely that transpiration demand and thus derived $S_r$-values are low. Figure 6 shows indeed a positive correlation between $S_r$ and leaf cover or tree height.

## 4.3 Vegetation types

Although not as strong as for the climate variables and the vegetation characteristics, relations between $S_r$ and vegetation types were found as well, especially for agriculture and pristine peatlands. A lack of strong patterns could, similarly as for the vegetation characteristics, for example be caused by the heterogeneity of the study catchments. The combined effect of different variables is another option that should especially be considered when looking at vegetation types. For example, when looking at the interaction between transpiration demand and vegetation type: does the existence of agriculture or deciduous forest increase transpiration rates and thus derived $S_r$-values, or are these vegetation types more likely to occur in areas with larger differences between water supply and demand? And linked to this, how large is the influence of the return period to which the vegetation adjusts: agriculture is likely to adjust to a shorter return period than forest. Or what is the role of soil: the used method assumes that soils are not important for the derived $S_r$, but they probably influence which vegetation will develop, which again influences the transpiration demands. Or how do the development of vegetation type and climate exactly coincide: especially peatland showed to be strongly correlated to climate (Figure 8), but to smaller extends agriculture and deciduous forest as well. To answer these questions, more detailed analysis of specific catchments would be required.

When looking especially at pristine peatlands it can be seen that they have a strong relation with the derived root zone storage capacity. In case of more than 20% pristine peatland cover, $S_r$ does not exceed the earlier found threshold of 115 mm. This may indicate that the "below threshold" conditions are ideal for the development of peat lands, which makes sense as peatlands develop in areas where precipitation exceeds evaporation and thus moisture conditions favour creation of peatland vegetation. In the developed peatlands generally the available space for root development is small, due to high groundwater

tables and fully saturated soil moisture conditions (e.g. Menberu et al., 2016). However, this is not explicitly accounted for in the $S_r$ calculations. This indicates that the pristine peatlands do not have a high transpiration demand and that evaporation is not excessively increased by high ground water tables. Typically evaporation from peat surfaces is small, especially if the water levels are below the growing sphagnum vegetation (Wu et al., 2010). Catchments where peatland is drained for forestry show

another pattern: the correlation with $S_r$ is lower, but especially the threshold seems to be weaker. The variation between the two groups for the threshold analysis is larger for pristine peatlands than for drained ones (Mann-Whitney U-test, p=0.0008 and p=0.0135 respectively). An effect could be expected since the motivation for artificial drainage is to create suitable soil moisture conditions for trees and increase forest growth (Sarkkola et al., 2013a). Peatland drainage has shown to have many effects on hydrological processes (ie. low flows, peak flows), which could partly be explained by the change in $S_r$.

Overall, the used data shows a variable relation between $S_r$-values and both vegetation characteristics and vegetation types in boreal landscapes. This is especially interesting as forestry actions together with shifting vegetation regions are moving towards the north (e.g., Hasper et al., 2016), which may thus result in different outcomes for root zone storage properties. Therefore it would make sense for future catchment scale studies focusing on the effects of changes in land use or climate on hydrological patterns, to take into account possible changes in $S_r$ as well.

## 4.4  Usefulness of a climate derived $S_r$

As shown in earlier studies, climate derived root zone storage capacities can be very useful in a modelling study. However, this study compared derived $S_r$-values with a set of catchment characteristics, which is a first step in exploring the wider application of $S_r$. The comparison with vegetation characteristics and types showed that the climate derived $S_r$ indeed also has some physical meaning in the study catchments. In addition, the comparison with climate variables showed that the (non-)coincides of

snow melt and the onset of potential evaporation has a large influence on the derived $S_r$-values. Combining these two findings, it can be expected that if the timing of either of them changes, the hydrological behaviour of boreal catchments can change remarkably. This finding for example may indicate that earlier snow melt decreases soil moisture during summer, resulting in larger root zone storage capacities. A possible increase in root zone storage capacity with increasing annual temperature and declining snow cover may cause also substantial changes to biogeochemical cycles (Wrona et al., 2016) and generated stream

flows (Bring et al., 2016). It would therefore be interesting to extend this research to other boreal and temperate regions. In such a study it can be investigated if the found threshold occurs in many areas with energy constrained evaporation or that it is mainly linked to the (non-)existence of snow cover.

With this in mind, a climate derived $S_r$ is especially valuable, as it will probably change when the climatic conditions (ie. amount of precipitation, snow-off date) or vegetation properties (ie. transpiration pattern) change. Before $S_r$-values can be

used in this way, more analyses should be carried out to investigate how (quickly) new equilibria are established and whether vegetation does change their survival mechanisms. However, when extending this line of thought, a climate derived $S_r$ can possibly be used to assess the hydrological effect of future changes in climatic and land cover conditions and the consequences for biogeochemical processes. This is essential in a global perspective, but especially in boreal regions which are facing drastic changes in near future resulting from joint pressures of intensified land use and climate change.

## 5 Conclusions

This paper showed that the climate based method to derive root zone storage capacities, with a snow component included, can be well applied to a range of boreal catchments. Subsequently, this paper investigated the relations between a set of catchment and vegetation characteristics and the derived root zone storage capacities to further understand the possibilities and physical meaning of this parameter. A climate derived $S_r$ was compared with climate variables, vegetation characteristics and vegetation types. A comparison between $S_r$ and the vegetation characteristics showed in general a positive correlation between $S_r$ and leaf cover, tree length and root biomass. This comparison had not been carried out before and further supports the plausibility of the climate-based method. Another important finding is that especially the (non-)coincidence of the snow-off and the onset of potential evaporation has a large effect on the derived $S_r$. In the studied regions, where evaporation is energy constrained, these two are the main variables determining the supply and demand of water. Further, it was observed that catchments with a large pristine peatland cover have small $S_r$-values and that for colder regions the influence of individual climate variables on $S_r$ is larger. A climate derived $S_r$ enables reflecting (changes in) climatic and vegetation conditions in a hydrological parameter. Therefore it gives additional information about the hydrological characteristics of an area and it could be beneficial to assess the effects of changing conditions.

*Data availability.* All data used for this study originates from open access databases which are listed in the data use subsection

*Competing interests.* The authors declare that no competing interests exist.

*Acknowledgements.* We would like to thank Maik Renner and two anonymous reviewers for their valuable comments: these really helped us to improve the manuscript.

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

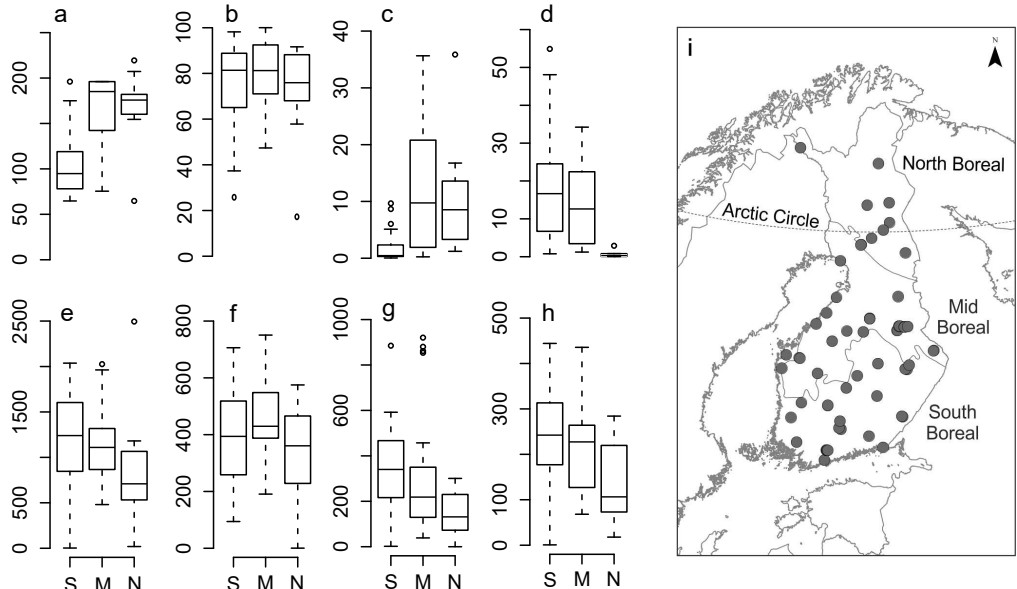

**Figure 1.** a) maximum snow water equivalent ($S_{SWE}$, mm), b) percentage forest (%), c) percentage of pristine peatlands (%), d) percentage of agricultural areas (%), e) total tree root biomass (10 kg/ha), f) pine root biomass (10 kg/ha), g) spruce root biomass (10 kg/ha), h) deciduous root biomass (10 kg/ha) at different ecoregions (S is south boreal, M is mid-boreal and N is north boreal).

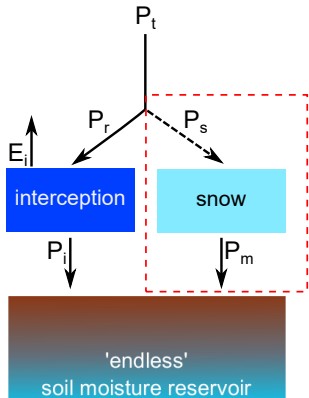

**Figure 2.** Schematisation of the method to calculate $S_r$, including snow module; the part in the red square is added for this research, the 'endless' soil moisture reservoir is similarly to the one in de Boer-Euser et al. (2016). The arrow for $P_s$ is dashed as this flux is not actually calculated, but $P_m$ is derived from the change in $S_{SWE}$.

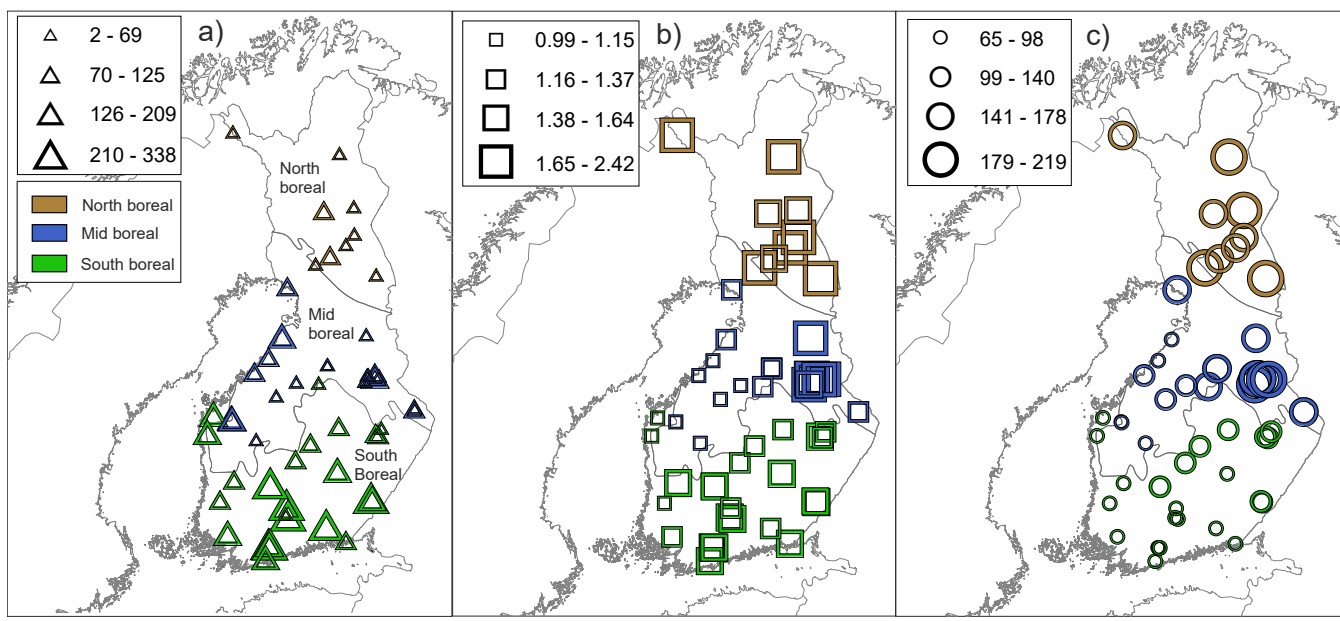

**Figure 3.** Map with study catchments and a) calculated root zone storage values ($S_r$, mm), b) ratio of precipitation and potential evaporation, and c) maximum snow water equivalent ($S_{SWE}$, mm). Different boreal ecoregions (south boreal, mid-boreal and north boreal) are shown in colors of the symbols and boundaries of ecoregions are marked with gray lines.

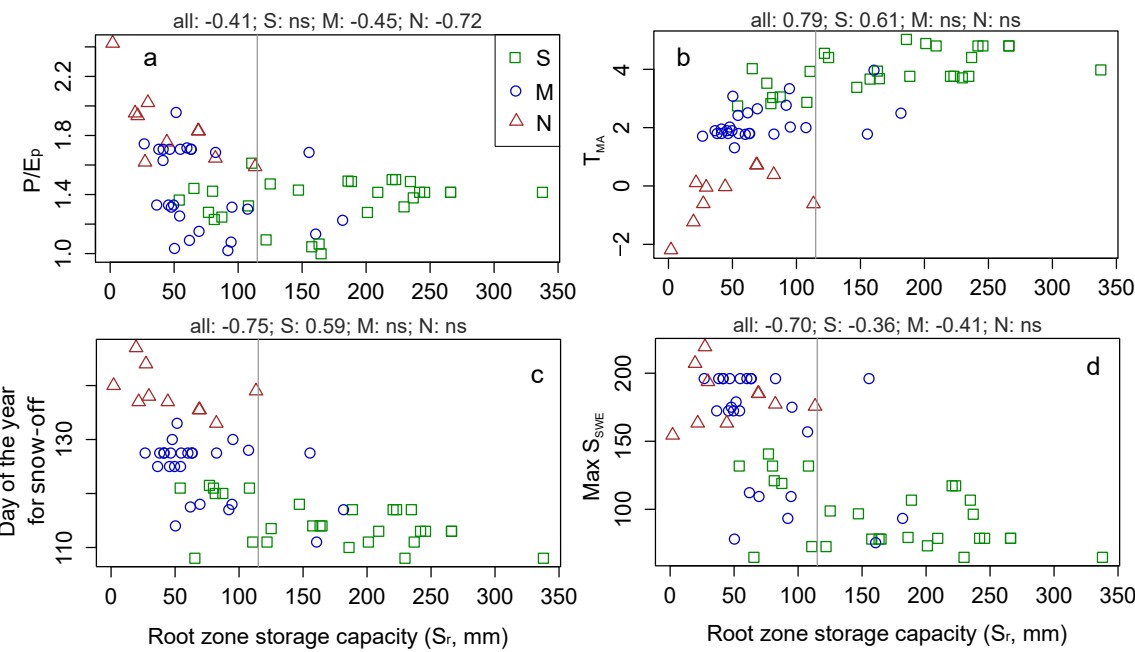

**Figure 4.** Root zone storage capacities and a) ratio of average precipitation and potential evaporation ($P/E_p$), b) mean annual temperature ($T_{MA}$), c) day of the year for snow-off, and d) maximum snow water equivalent ($S_{SWE}$) in the catchment at different ecoregions (S is south boreal, M is mid-boreal and N is north boreal). The titles of the subplots show the Spearman's correlation coefficients (significant correlation for p<0.05). The line at 115 mm illustrates the discussed threshold.

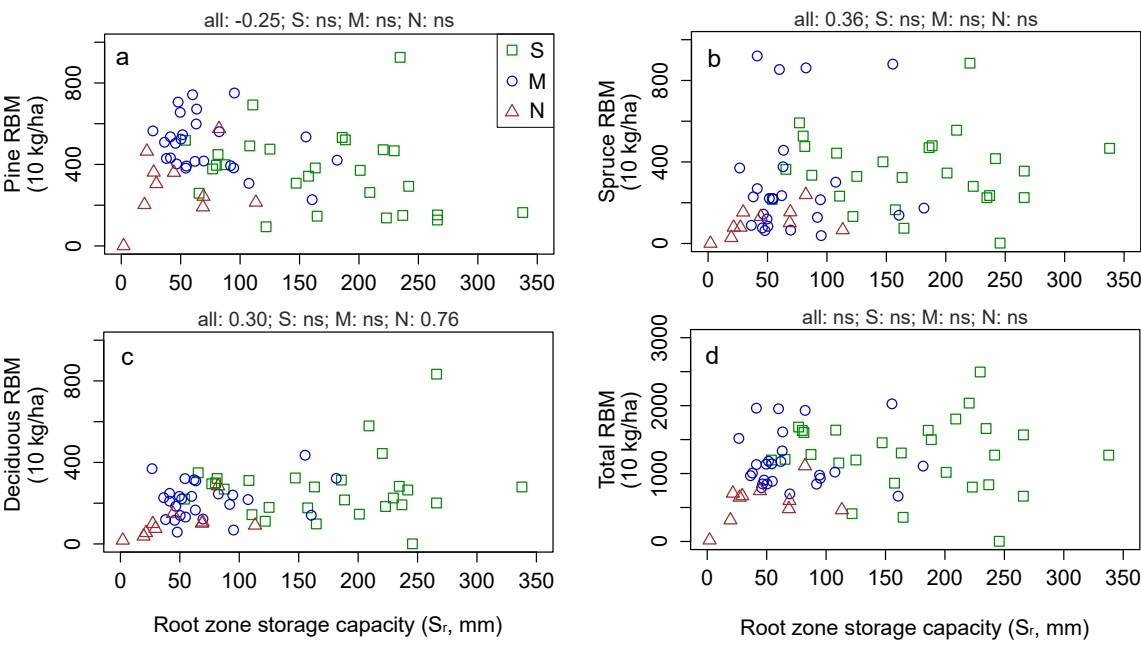

**Figure 5.** Root zone storage capacities and a) pine root biomass (RBM, 10 kg/ha), b) spruce RBM (10 kg/ha), c) deciduous RBM (10 kg/ha) and d) total RBM (10 kg/ha) in the catchment at different ecoregions (S is south boreal, M is mid-boreal and N is north boreal). The titles of the subplots show the Spearman's correlation coefficients (significant correlation for $p < 0.05$).

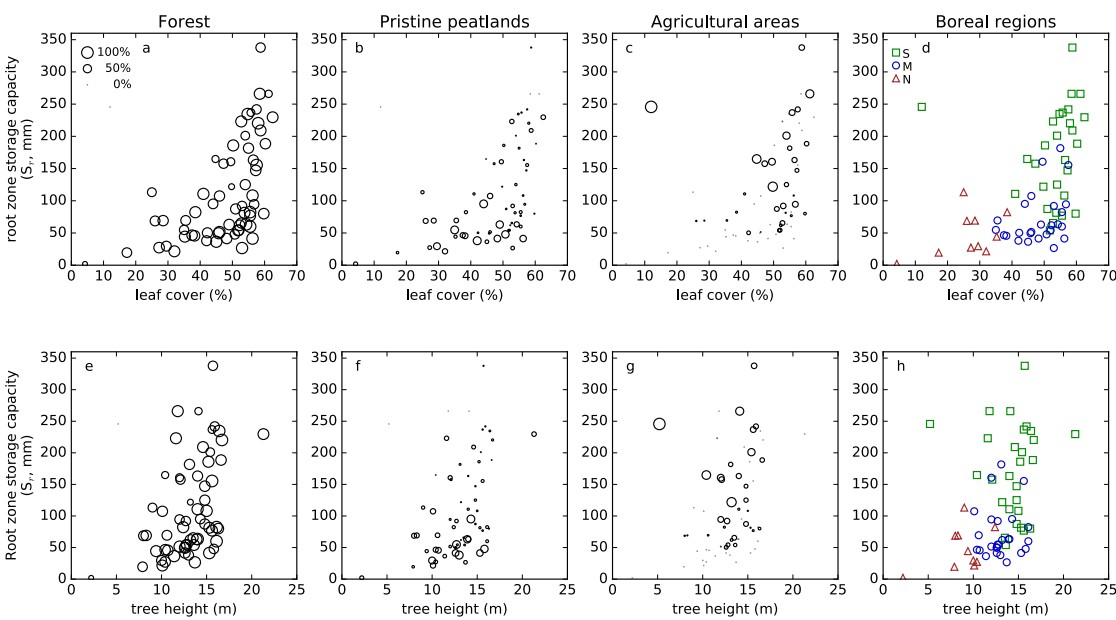

**Figure 6.** Calculated root zone storage capacity versus average leaf cover (top) and tree height (bottom) of four years. Larger circles indicate higher percentage of vegetation type for a&e) forest, b&f) pristine peatlands, c&g) agriculture; d&h) are colour coded by boreal region. $S_r$ has statistically significant Spearman's correlation with leaf cover ($r = 0.33$) and tree height ($r = 0.32$). Different boreal regions did not resulted in statistically significant correlations when considered individually.

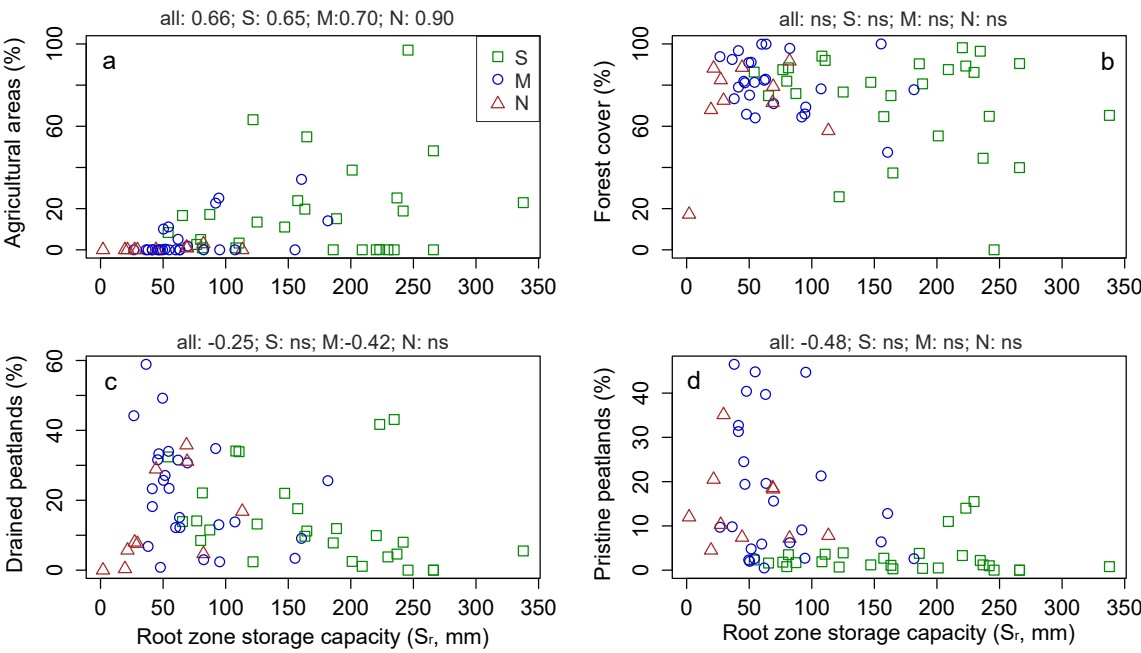

**Figure 7.** Root zone storage capacities ($S_r$, mm) and proportion of a) agricultural areas (%), b) forest cover (%), c) drained peatlands (%) and d) undrained peatlands (%) in the catchment at different ecoregions (S is south boreal, M is mid-boreal and N is north boreal). The titles of the subplots show the Spearman's correlation coefficients (significant correlation for p<0.05).

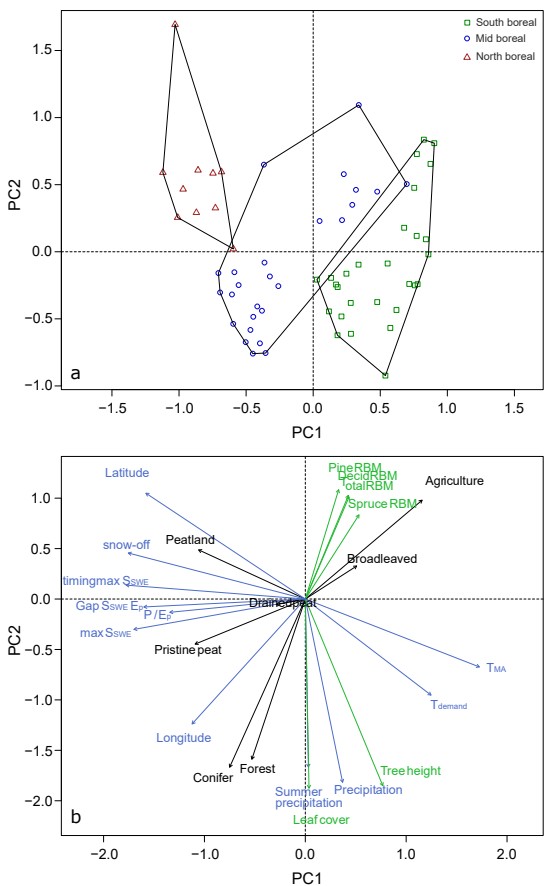

**Figure 8.** Principal component analysis with the catchment characteristics that are being compared with $S_r$ in the study. a) Catchments plotted on PC1 and PC2, with boreal regions indicated. b) Catchment characteristics with their loadings on PC1 and PC2; catchment characteristics are divided into three categories: climate (blue), vegetation characteristics (green) and land use types (black). Note that for readability the axis of the two plots are not the same.

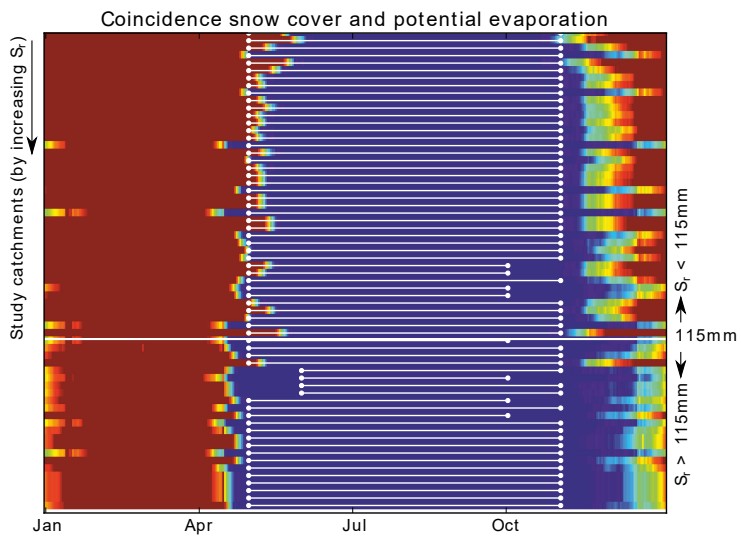

**Figure 9.** Snow cover is presented by the colour plot (red: $S_{SWE} > 15$ mm, blue: $S_{SWE} = 0$). Occurrence of potential evaporation ($E_p > 0$) is presented by white lines; note that the actual amount of $E_p$ is not presented. Presented data are long term daily averages. Catchments are ordered by increasing $S_r$-values.