# Peer review of "Understanding variability in root zone storage capacity in boreal regions"

_Hydrology and Earth System Sciences, 2018_

## Referee Comment (RC1) · Anonymous Referee #1 · 20 Apr 2018

Overall Comments:

The authors have produced a nice study that attempts to find correlations between a climate derived root zone water storage (Sr) and aspects of vegetation distribution among 64 drainage basins in Finland. This topic is of interest to the readers of HESS and the general findings are useful and well presented. My recommendation is that revisions are necessary but that the potential for eventual publication is strong.

A weakness in the analysis is that Sr is derived at the basin scale, and then assessed against vegetation type and attributes, but different vegetation types prefer different soil texture and moisture conditions. A diverse catchment likely has a diverse soil and an Sr value that may not apply very well to any of the vegetation types in the particular catchment. Conversely, I would expect stronger correlations and more valid

Sr values in catchments containing a dominant vegetation type. From looking at Figure 3, I suspect that peatlands never make up enough of the catchments for the Sr value obtained to be applicable to them, and this would also be the case for many of the agricultural areas. Forest is usually the dominant vegetation type and this is shown by its close agreement with the broad boreal zone plot.

There appear to be inconsistencies in the data that are presented in subsequent plots. These need to be corrected or explained. In Figure 3a there are two points with relatively low leaf cover and Sr values of about 230 and 110 mm. These points show up in the boreal regions plot (3d) as northern points, so they are northern forests. Based on Figures 3e and 3h, the 230 mm Sr value is also associated with a large tree length of about 210 m while the 110 mm Sr value is associated with a medium tree length of about 100 m. I'm wondering if there is something about these two basins that makes them different. Why is the leaf cover low and yet one of them has the largest observed tree length? Was there a defoliation event? When I look in Figure 4, I see no northern forests with an Sr value anywhere near 230 mm. At first I thought that perhaps it wasn't pine, spruce or deciduous, but it doesn't even appear in figure 4d. What happened to this forest that stands out in Figure 3? Figure 3 shows two northern catchments with Sr values not far from 110 mm, one is mostly forest with some peatland and the other has a bit less forest but still more forest than peatland. Figure 4 only shows one northern catchment with Sr values close to 110 mm. What happened to the other catchment? Figure 3 shows two northern catchments with Sr of 70-80 mm and one at about 50 mm, and these appear to be forests or mostly forests, but in Figure 4 the Sr values do not fit the same distribution of two in the 70-80 mm range and one at 50 mm but instead it appears that two are at about 70 mm and one at about 85 mm. These plots appear to be derived from somewhat different datasets with respect to Sr values. The data used in the figures needs to be made internally consistent, or explanations provided for data appearing in some figures and not in others.

From my experience, some pine species like to grow in sandy well-drained soil, and

here contribution to discharge is likely high and transpiration low. In such a catchment the estimated T should be low and there will not likely be large deficits, even though the soil can get quite dry. Spruce trees like to grow in moist soil, often in poorly drained areas. Such areas don't often dry out and contributions to discharge also likely follow precipitation quite well, except following a drought when there is recharge; again such areas may not see very large deficits. So we have pine in dry areas with small deficits and spruce in wet areas with usually small deficits. Deciduous trees tend to have larger transpiration demands and can grow in poorly or well drained soils. If deciduous trees exist more often in areas with larger deficits and adjust their root mass accordingly, this may explain why the best correlation is for deciduous trees in Figure 4. However, much of this detail would be smeared out because each basin contains multiple tree and other vegetation types and probably a combination of wet and dry areas. With this in mind, I understand why the correlations and patterns are not as strong as one might hope for.

Some of the relationships appear to be curvilinear rather than linear, so it might be more informative to try fitting some nonlinear relationships (exp, log, polynomial) to see which correlations increase and whether the relative importance of parameters changes. Perhaps a flexible generic nonlinear model could be used.

More specific comments:

P4 line 4-7: Are the authors aware of the type of precipitation gauges used to measure snowfall, whether they were shielded and whether they were corrected for undercatch based on coincident wind speed measurements? Precipitation gauges always measure less than the true snowfall amount, but if properly located, shielded and adjusted using established correction factors based on wind speed, one can arrive at an accuracy that is comparable with a snow survey.

P4 line 22-23: I am somewhat perplexed that canopy interception is included for rain but not for snow, when it is well known that boreal forests can store close to an order

of magnitude more mass of snow versus water on the canopy, and interception losses on the order of 30% or more are common over a winter.

P5 Section 2.4: I think an explanation of the specific method used to obtain Sr is required. I looked at de Boer-Euser et al. (2016) and based on that, I think I understand what was done, but a brief overview would be helpful.

In Figure 3 are the values of leaf cover and tree length basin values or are they specific to each vegetation type? I see for example the two northern basins with Sr near 70-80 mm and leaf cover near 24-28% in Figure 3d, and these appear to have corresponding large forest fractions, small peatland and smaller agricultural fractions with the same leaf cover values. This suggests that these values are basin-scale and are not specific to each vegetation type. Since most of the basins are forest-dominated, when we look at the peatlands or agricultural plots, in most cases when the fractions of these vegetation types are small, we are not looking at leaf cover or tree length values that have anything to do with the peatland vegetation or crops other than they happen to be in the same basin. This should be made more clear.

P6 line 19: The statement "...and this correlation decreases for higher percentages of peatland..." is a bit misleading. There hasn't really been an analysis of correlation for basins with high and low peatland cover. When I look at Figure 3f, it does appear that there may be some correlation between Sr and tree length in pristine peatlands for the basins with small fractions of pristine peatlands (because the correlation is coming from the larger forest fractions) and the pattern looks more scattered (implying a lower correlation) for the larger circles or basins with a larger peatland fraction. It should be made clear that these are just visual interpretations, not a comparison of calculated correlation coefficients.

P6 line 20: The variability in leaf cover and tree length is small within the boreal regions but appears greater when the three regions are examined together. It appears that factors affecting tree length and leaf cover act largely but not exclusively along the
latitudinal gradient such that the correlation is weak within each region.

I think the strong relationship between Day of Year (date of snow-off) and Sr has more to do with the fact that the snow-off date is correlated with both maximum SWE and air temperature than a special relationship with the phase of snowmelt. For example, the timing of maximum SWE is probably determined almost exclusively by temperature, whereas the amount of maximum SWE is a combination of snowfall amount and temperature (and other factors).

P7 line 22: While it is true that the clearing of land for agriculture increases soil exposure (more evaporation) and crops tend to have high transpiration demands (more transpiration), there is also the likelihood that croplands are more prevalent in the south because of the longer growing season and increased likelihood of a successful crop. So did the crops in the south cause larger Sr values because of their higher water demands or were they planted in a warmer area because it is beneficial for the crops and that just happens to coincide with larger Sr values (warmer, more evaporative demand)? I would say it works in both directions.

P8 lines 1-3: Peatlands generally develop in areas where the soil does not dry out very often, either because of cold temperatures and low evaporative demand, or a combination of positive P-E and poor drainage. Since the soil does not tend to dry out, the Sr value calculated will be small because large deficits of P-T are rare.

P8 line 11-12: Maximum SWE and mean annual temperature and the snow off date are likely highly correlated within a small region. A regression model that attempted to include all three would almost certainly show that all three are not necessary. I would be inclined to predict that mean annual temperature and maximum SWE are the most important, but maximum SWE is partially dependent on mean annual temperature based on the length of the snow period and when melt starts. Perhaps mean annual temperature and winter precipitation would do better.

P9 line 6: I have read that jack pine have a tap root to access deeper water. If this is

true of the pines in the Finland catchments, it may be that deeper water is accessed without a large increase in root density and this may lessen correlations between Sr and root biomass. The authors would want to find an appropriate citation before using this point as an explanation.

P9 lines 1-3: Yes, peatlands develop in places where the decomposition rates are slower than the annual increment, due to a combination of cold temperatures and/or poor drainage and anoxic conditions. Peatlands are created by the same conditions that cause the estimated Sr to be low, but I doubt that peatlands cause the small Sr values.

Minor comments and corrections:

Sr,20 is never defined in the text. It is stated in Section 2.3 that a drought return period of 20 years is used, but the symbol Sr,20 is not introduced here; it simply appears in figures but not in the text.

P4 line 21 and elsewhere: Why is SSWE used for Snow Water Equivalent instead of SWE?

P5 line 14: "Tree length" is never defined. It is certainly not tree height, but I don't see the term in the literature.

P7 line 24-25: In Fig. 7c I might view the southern boreal region as showing a negative correlation between Drained peatland % and Sr with two outliers.

P9 line 24: I would change ". . .. for example indicates that. . .." to ". . . for example may indicate that. . .."

Figure 1: Add a North Arrow. Perhaps outline Finland so as to make the study area boundaries more clear.

Figure 3: The letters need to be on the plots (e.g. a, b, c . . .. h).

Figure 6: Change Julian date to Day of Year. Julian date or Julian day is not the same

as Day of Year.

Figure 8: What do the size of the boxes represent? There is no scale provided to interpret this.

[Figure]

---

## Referee Comment (RC2) · M. Renner (Referee) · 8 May 2018

The manuscript by de Boer-Euser presents an application of root zone storage (S) derived from hydro-climate data for the boeral region of Finland. I am in favor of the idea that climate, soil and vegetation are co-evolving and that the plant available water storage is an outcome of this interdependence and not just a fixed parameter determined simply from soil / vegetation lookup tables. In boreal regions we also expect that snow cover, high water tables, low soil depths, and frozen soils are important constraints on root zone storage. de Boer-Euser explicitly implement this through a snow component driven by observed changes in snow water equivalent. The other constraints are not explicitly modeled but somehow implicitly included through the observed long term water balance closure ($E = P − R$) which is an input to the method – if I understand this

correctly.

I really like the motivation to compare the derived S with independent data, such as forest characteristics and land cover type. The results show that S increases towards the south, increases with biomass, but decreases with area of peatlands due to high water tables. Since root biomass is used as a metric for verification, more details are required on how this was derived.

The statistical analysis is presented in a way to suggest that root zone storage is independent of the climate variables (P, Ep, T, SWE), while indeed it is derived from these data. Actually the analysis of climate controls is performed like an uncontrolled sensitivity analysis of a bucket model with different inputs. The outcomes of this sensitivity analysis (Fig 6, Sect. 3.2, 4.2) are difficult to interpret since the influence of the other input parameters changes from one catchment to the next. I also wonder why there is no precipitation frequency / drought index be used to correlate with S?

An interesting point is the influence of drainage of peatlands on S. Although the authors claim to identify an effect, I could not identify the mentioned influence of drainage in Fig 7. Unfortunately, the analysis lacks a reference to compare drainage with pristine peatlands. Here a stratification of the data could be useful means to assess this point.

I do not understand how the method can be applied in climate or land-use change analysis. To my understanding an estimate of transpiration is required to estimate S and both are unknown for a given change scenario. Please explain.

Apart from these major remarks the authors did a very good job in writing and presentation of the results. The topic is of high scientific interest and after revisions well suited for publication in HESS.

Detailed comments:

P1L17: Check the causal order of the mentioned processes "Retreating . . . "

P2L2: add references

P2L7: "but so far none have studied changes in transpiration (patterns) at the catchment scale in boreal regions." Please check (Jaramillo et al., 2018; van der Velde et al., 2013).

P2L17: "Thus, climate (or the balance between precipitation and transpiration) has a large influence on the developed $S_r$." Doesn't transpiration depend on the root zone storage (and not the other way around)?

P4L3ff: To my knowledge there is a significant undercatch of precipitation, especially in winter. It is not clear if the undercatch was corrected for, but if not, then I disagree with the choice of the authors to correct SWE with P.

Sect 2.3 climate derived root zone storage capacity. Since the results show how climate variables correlate with S, I recommend to repeat the key equations to show how climate input is used in the method. Then also the choice of a return period of 20yr may become more clear.

P5L12: wording "transpiration demands" is unclear to me

Results / Discussion: report correlation and significance in text. For example in Sect. 3.1

P8L21: check argument: "The presented results show that climate derived root zone capacities are related to vegetation characteristics, climate variables and vegetation cover, which strongly indicates that the $S_r$-method can be used for boreal regions containing seasonal snow cover." Since S is computed from climate data, the relationship is not a verification of the method!

P9L2: unclear from results "This seems to indicate that in case of low transpiration demands the plant's resources between below and above soil elements are more equally divided than for areas with higher transpiration demands."

P9L6f: unclear argument "However, for pine in mid- and south-boreal regions a negative correlation was observed, which means that the vegetation is able to create a

larger storage capacity with fewer or thinner roots." Please calculate the significance of the correlation and possibly use a bootstrap to check the influence of outliers. Please check/report how root biomass was calculated. Also check for other influencing variables.

P9L12: please provide references for shifting management activities

P10L5: please provide references

P10L8: "Peatland drainage for forestry changed this pattern: higher S r values were observed in areas with larger cover of drained peatlands (Figure 7)." I could not see this effect!?

Sect. 4.4: Explain how the method is applied to a change scenario when data on transpiration is required a priori?

Figure 1: Missing y-axis labels; Add points to the boxplots. Panels of Fig1 are insightful, but hardly touched in text. Add relevant topographic info to the map.

Fig 2: use white text in dark boxes

Fig 6c, Fig 8: Julian date for snow off in Fig.6 and Julian Date for max SWE in Fig 8. Please be consistent.

Fig 7c,d: Peatland area per catchment? Why does the number of points change?

Fig 8: show correlation as text in one of the diagonals

Fig. 9: What is the ordering in y-axis? Coloring: black lines are hardly seen on darkblue background. Why is PET always the same?

References:

Jaramillo, F., Cory, N., Arheimer, B., Laudon, H., van der Velde, Y., Hasper, T.B., Teutschbein, C., Uddling, J., 2018. Dominant effect of increasing forest biomass on evapotranspiration: interpretations of movement in Budyko space. Hydrol Earth Syst

[Figure]

Sci 22, 567–580. https://doi.org/10.5194/hess-22-567-2018

van der Velde, Y., Vercauteren, N., Jaramillo, F., Dekker, S.C., Destouni, G., Lyon, S.W., 2013. Exploring hydroclimatic change disparity via the Budyko framework. Hydrol. Process. https://doi.org/10.1002/hyp.9949

---

## Short Comment (SC1) · 28 May 2018

[supplement omitted: unrelated document]

---

## Short Comment (SC2) · 28 May 2018

Dear Maik Renner,

Thank you for the review of our manuscript and the interest in our study. Your comments made us realise that some elements are not yet explained well and that our argumentation misses some intermediate steps. Therefore they are very valuable for improving our manuscript. Below we have replied in more detail to all your comments.

*The results show that S increases towards the south, increases with biomass, but decreases with area of peatlands due to high water tables. Since root biomass is used as a metric for verification, more details are required on how this was derived.*

We will add a description for the calculation of root biomass. Data is based on multi-source national forest inventory data provided by Finnish Natural Resources Institute (LUKE). Data is based on field data, satellite images, digital map data and other georeferenced data sets. More information can be found from Mäkisara et al. 2016. http://jukuri.luke.fi/handle/10024/532147

*The statistical analysis is presented in a way to suggest that root zone storage is independent of the climate variables (P, Ep, T, SWE), while indeed it is derived from these data. Actually the analysis of climate controls is performed like an uncontrolled sensitivity analysis of a bucket model with different inputs. The outcomes of this sensitivity analysis (Fig 6, Sect. 3.2, 4.2) are difficult to interpret since the influence of the other input parameters changes from one catchment to the next. I also wonder why there is no precipitation frequency / drought index be used to correlate with S?*

We agree with you that the root zone storage capacities are dependent on climate variables. Actually, for the calculation of $S_r$ four climate parameters are used, namely: daily precipitation, daily snow water equivalent, long term averaged discharge and long term monthly averaged potential evaporation. Although climate parameters were used in Figure 6, these are not variables that are directly used in the calculations (mean annual temperature, maximum snow water equivalent, snow off date and the ratio of precipitation and potential evaporation).

As the estimation of $S_r$ is not one calculation, but derived from the simulated soil moisture deficit (we will explain this clearer in the description of the method) the influence of different climate variables is not always straight forward. Therefore, we used these plots to see if there is any correspondence between $S_r$ and climate variables that were not directly used in the estimation of $S_r$. We realise however, that we made this aim not fully clear in the discussion of the results and we will change the text accordingly in the revised version of the manuscript.

With regard to the precipitation frequency/drought index: we included $P/E_p$ in the analysis, which is a definition of the aridity index. Further, we compared $S_r$ with runoff coefficients during the analysis, which showed a strong relation. This is logical, as it is one of the main inputs in the estimation of $S_r$ and probably even has a stronger effect on the calculation than $S_{SWE}$; therefore we considered the other plots more interesting to incorporate.

With respect to precipitation frequency a comparison with inter-storm duration ($I_{isd}$) could be made; we did not do this during the first analysis. The relation between $S_r$ and $I_{isd}$, based on total precipitation and an interception capacity of 1.5 mm, can be seen in the figure below. It can be seen that the variability in $I_{isd}$ between the catchments is very limited. Therefore, we don't think that adding this plot in the revised manuscript is valuable for the analysis.

[Figure]

**Figure 1| Average interstorm duration in relation to derived root zone storage capacities, different symbols indicate different boreal regions (green squares = south boreal; blue circles = mid boreal; red triangles = north boreal)**

*An interesting point is the influence of drainage of peatlands on S. Although the authors claim to identify an effect, I could not identify the mentioned influence of drainage in Fig 7. Unfortunately, the analysis lacks a reference to compare drainage with pristine peatlands. Here a stratification of the data could be useful means to assess this point.*

From Figure 7 it can be seen that more catchments exist with larger $S_r$ values and larger percentages covered with drained peatland than with larger $S_r$ values and larger percentages covered with pristine peatlands. In Figure 8 a negative correlation between $S_r$ and both drained as pristine peatlands can be observed, although this correlation is stronger for pristine peatlands. However, two reasons for this difference can be given. First, the drainage of peatlands for forestry probably creates larger transpiration demands and thus larger root zone storage capacities. Or, second, as most of these drained catchments are located in the south boreal region, it can also be that $S_r$ values were already higher in these catchments before the peatlands were being drained. Unfortunately the available data series are not long enough to compare $S_r$ values before and after drainage of some of the drained catchments. In the revised version of the manuscript we will change the text accordingly to make this clearer.

*I do not understand how the method can be applied in climate or land-use change analysis. To my understanding an estimate of transpiration is required to estimate S and both are unknown for a given change scenario. Please explain.*

For climate and land-use change analysis often data are available for a long period containing a change, this change is probably reflected in the corresponding $S_r$ values as well. In case change scenarios are used, these scenarios can include a change in precipitation and/or discharge and (additionally) a (relative) change in transpiration. With respect to the land-use change analysis, this can include a hypothesised change in transpiration as well. In addition to this, it is likely that different vegetation types adjust to different return periods (Wang-Erlandsson et al, 2016), so a change in land-use can in that way have an effect on the estimated $S_r$ values.

In our opinion it may be worthwhile to include the estimation of (changed) $S_r$ values in these types of analyses, as they could give more information about how the hydrology of a catchment changes under the studied climate or land-use scenarios.

We will elaborate this in more detail in the revised manuscript.

Detailed comments:

*P1L17: Check the causal order of the mentioned processes "Retreating..."*

We will modify the sentence accordingly.

*P2L2: add references*

We will add support references, for example

Laudon et al. 2011. Consequences of More Intensive Forestry for the Sustainable Management of Forest Soils and Waters. Forests 2, 243-260.

Nieminen et al. 2017. Impacts of forest harvesting on nutrient, sediment and dissolved organic carbon exports from drained peatlands: A literature review, synthesis and suggestions for the future. Forest ecology and management 392, 13-20.

*P2L7: "but so far none have studied changes in transpiration (patterns) at the catchment scale in boreal regions." Please check (Jaramillo et al., 2018; van der Velde et al., 2013).*

We will change the sentence and consult/incorporate the mentioned references.

*P2L17: "Thus, climate (or the balance between precipitation and transpiration) has a large influence on the developed $S_r$." Doesn't transpiration depend on the root zone storage (and not the other way around)?*

Yes, we agree with you that transpiration is sustained by the root zone storage capacity and in that sense influenced by it. This is reflected in the used method by assuming that the vegetation has developed a root zone storage capacity to sustain the transpiration demands. However, to calculate this required root zone storage capacity, the long term water balance is used to estimate the transpiration demands of the vegetation. This entire approach assumes equilibrium in the catchment and therefore can be seen as working in two directions: if either the root zone storage capacity or the transpiration demand changes, the other will (probably) change as well. We will make this clearer in the extended description of the method (see also our reply to the comments of the first reviewer).

*P4L3ff: To my knowledge there is a significant undercatch of precipitation, especially in winter. It is not clear if the undercatch was corrected for, but if not, then I disagree with the choice of the authors to correct SWE with P.*

We used a spatially interpolated dataset with a resolution of 10 x 10 $km^2$ for the meteorological parameters (precipitation, air temperature) constructed by Finnish meteorological institute (FMI). In this data set the measurement error caused by gauges has been checked and corrected in operative quality control. For snow data ($S_{SWE}$), we used snow line data provided by Finnish Environment Institute and measured by standard methods. Since $S_{SWE}$ was closest available and not always situated within the study catchment, we corrected $S_{SWE}$ with local precipitation.

*Sect 2.3 climate derived root zone storage capacity. Since the results show how climate variables correlate with S, I recommend to repeat the key equations to show how climate input is used in the method. Then also the choice of a return period of 20yr may become more clear.*

We will include more details about the used method in the revised manuscript (see also our reply to the first reviewer). The choice of a 20 year return period follows from the analysis of Gao et al. (2014), who found that on average it is most likely that vegetation adapts it root system to a drought with a return period of 20 years.

*P5L12: wording "transpiration demands" is unclear to me*

Transpiration demand is used for the long term deficit between precipitation and discharge. The vegetation in the catchment should be transpired this about of water to close the long term water balance with the given precipitation and discharge. We will make this clearer in the extended description of the method to estimate $S_r$.

*Results / Discussion: report correlation and significance in text. For example in Sect. 3.1*

Where relations between variables are discussed in the text, we will add correlation coefficients and significance as well.

*P8L21: check argument: "The presented results show that climate derived root zone capacities are related to vegetation characteristics, climate variables and vegetation cover, which strongly indicates that the S r - method can be used for boreal regions containing seasonal snow cover." Since S is computed from climate data, the relationship is not a verification of the method!*

We agree with you that this statement is not well formulated; the relationship between climate data and variation of $S_r$ values is indeed not a verification of the method. However, as discussed earlier, we think it valuable to incorporate the comparison with some climate variables. In the revised manuscript we will change the argument into: "The presented results show that climate derived root zone capacities are, besides climate variables, also related to vegetation characteristics and vegetation cover."

*P9L2: unclear from results "This seems to indicate that in case of low transpiration demands the plant's resources between below and above soil elements are more equally divided than for areas with higher transpiration demands."*

In Figure 3a stronger relation between leaf cover/tree length (above ground biomass) and derived $S_r$ values (below ground biomass) can be visually observed for smaller $S_r$ values (< 115 mm). For larger $S_r$ values, the leaf cover/tree length is more constant. This indicates that the vegetation uses more resources for below ground biomass in cases of larger $S_r$ values. As the derived $S_r$ values are strongly determined by the transpiration demands, the catchments with large $S_r$ values also have high transpiration demand.

We will change the text in the revised manuscript accordingly.

*P9L6f: unclear argument "However, for pine in mid- and south-boreal regions a negative correlation was observed, which means that the vegetation is able to create a larger storage capacity with fewer or thinner*

*roots."* *Please calculate the significance of the correlation and possibly use a bootstrap to check the influence of outliers. Please check/report how root biomass was calculated. Also check for other influencing variables.*

We will report in more details the calculation method for root biomass. We will look into correlations and incorporate the outcome in the revised manuscript.

*P9L12: please provide references for shifting management activities*

We will add some references in the revised manuscript for example: Hasper et al. 2016. Water use by Swedish boreal forests in a changing climate, Functional Ecology 30, 690-699.

There is no specific reference for shifting management activities. However, when forest resources are growing faster due to changing climate also forest management activities shift.

*P10L5: please provide references*

We will add some references in the revised manuscript to support statement for peatlands and high water tables. For example: Menberu M, Tahvanainen T, Marttila H, Irannezhad M, Ronkanen A-K, Penttinen J, Kløve B. 2016. Water table-dependent hydrological changes following peatland forestry drainage and restoration: Analysis of restoration success. Water Resources Research, 52(5), 3742-3760.

*P10L8: "Peatland drainage for forestry changed this pattern: higher S r values were observed in areas with larger cover of drained peatlands (Figure 7)." I could not see this effect!?*

As discussed earlier, there can be two reasons for the difference between drained and pristine peatlands. We will change the text of the revised manuscript accordingly to make this clearer.

Sect. 4.4: Explain how the method is applied to a change scenario when data on transpiration is required a priori?

Data on transpiration is indeed used in the analysis; however, this data is derived from the long term water balance (precipitation, discharge and $S_{SWE}$). When the change scenarios are constructed, the transpiration can again be estimated from the water balance or be assumed to change in a certain way. By subsequently calculating $S_r$ values, the effect on the hydrology of the changing conditions can be further explored. We agree that this is not yet a complete analysis, but we definitely see a potential for further research.

We will change the text to make this clearer.

*Figure 1: Missing y-axis labels; Add points to the boxplots. Panels of Fig1 are insightful, but hardly touched in text. Add relevant topographic info to the map.*

We will change the figure accordingly; however, adding topographical info will make the figure less clear to read, so we prefer not to include it.

*Fig 2: use white text in dark boxes*

We will change this in the revised version of the manuscript

*Fig 6c, Fig 8:  Julian date for snow off in Fig.6 and Julian Date for max SWE in Fig 8. Please be consistent.*

We will change all these to 'day of the year', as suggested by the first reviewer.

*Fig 7c,d: Peatland area per catchment? Why does the number of points change?*

Figure 7c illustrates the percentage of the catchment covered with drained peatlands and Figure 7d shows the same for pristine peatlands. The Number of points change since some catchments does not have pristine peatland areas and vice versa.

*Fig 8: show correlation as text in one of the diagonals*

We preferred to only mention the correlation values in the text to prevent the figure from overflowing with information. For the revised manuscript we will see if there is a way to include the values in the figure as well.

*Fig. 9: What is the ordering in y-axis? Coloring: black lines are hardly seen on dark blue background. Why is PET always the same?*

The ordering on the y-axis is by increasing estimated $S_r$ value and the figure does not show the amount of potential evaporation, but the period in which $E_p$ is occurring/measured. We will clarify this in the figure caption and change the colour of the lines to white.

References:

Jaramillo, F., Cory, N., Arheimer, B., Laudon, H., van der Velde, Y., Hasper, T.B., Teutschbein, C., Uddling, J., 2018.  Dominant effect of increasing forest biomass on evapotranspiration:  interpretations of movement in Budyko space.  Hydrol Earth SystSci 22, 567–580. https://doi.org/10.5194/hess-22-567-2018

van der Velde, Y., Vercauteren, N., Jaramillo, F., Dekker, S.C., Destouni, G., Lyon, S.W., 2013.  Exploring hydroclimatic change disparity via the Budyko framework.  Hydrol. Process. https://doi.org/10.1002/hyp.9949

Gao, H., M. Hrachowitz, S. J. Schymanski, F. Fenicia, N. Sriwongsitanon, and H. H. G. Savenije (2014), Climate controls how ecosystems size the root zone storage capacity at catchment scale: Root zone storage capacity in catchments, Geophys. Res. Lett., 41, 7916–7923, doi:10.1002/2014GL061668.

Wang-Erlandsson, L., Bastiaanssen, W. G. M., Gao, H., Jägermeyr, J., Senay, G. B., van Dijk, A. I. J. M., Guerschman, J. P., Keys, P. W., Gordon, L. J., and Savenije, H. H. G.: Global root zone storage capacity from satellite-based evaporation, Hydrol. Earth Syst. Sci., 20, 1459-1481, https://doi.org/10.5194/hess-20-1459-2016, 2016

---

## Referee Comment (RC3) · Anonymous Referee #3 · 5 Jun 2018

The study is on an important and interesting topic, as little is known about potential changes in catchment storage properties under climatic and land use change. Potential to improve our hydrological predictions under climatic and land use change is limited by lack of information and understanding, so studies in this area can be expected to be in demand by HESS audience. Another strength of this study is the dataset, which is very well described and referenced. The article is nicely structured, and the results are presented well (though I would prefer to see more numerical information to back up some claims).

My main concern about this study is about how much of the results originate from self-prediction given the high correlation between source and comparison data. Sr is derived based on climatic records, and then Sr is compared with climatic and vegeta-

tion properties which are known to be related with the source data Sr was derived from. The authors do acknowledge the relationship (e.g. p 5 l 24-25, p 6, l 2-3, p 9 l 16-17), but they still interpret results in a way where (higher) correlation implies control over Sr, which I think is questionable. The results might reflect just the closer correlation with the source climatic data for Sr, and not causal relationship with the soil storage properties. For example, it remains unclear to what extent the relationship between Sr and vegetation properties/land use are just a consequence of both being related to the climate. In this case the change in vegetation would not influence Sr as it can be expected from the results, if the vegetation is the only thing changing. This would particularly apply to cases where the results are somewhat counterintuitive (e.g. p6 l26-27 and Fig 4a, or p7 l23-25 and Fig 7b where decrease in forested area is associated with increase in Sr).

In this light, I think it would be more informative and would give more confidence in the results to apply some method which can account for a number of potential "controls" and assess their importance against each other, for example PCA or multimodel inference (e.g. Saft et al, 2016).

I am also a bit puzzled about the gap between snowmelt and onset of PET, as both are governed by exactly the same increasing energy flux (temperature/sunshine). I would assume that this gap should be very closely related to the maximum SWE ($\sim$more snow takes longer to melt). Anyway, it would be interesting to calculate this gap (using some threshold for snowmelt) and include it directly as yet another factor along with the other characteristics used. I wonder why it was treated separately.

On a different note, it would be good to see more numerical information (i.e. Spearman's rho, and associate p value) associated with positive / negative correlations described in the text. It is difficult to extract relevant information from figure 8, especially since it is not numeric. Fig 8 also does not include correlation results for sub-regions which are mentioned in the text, and I could not find these results anywhere else.

Importance and implications:

What is the use of the derived Sr and discovered relationships with other characteristics? And in the context of climate change, would not it be easier to derive new Sr following the original method accounting for climate change in the source data instead of looking at the correlations?

Specific comments:

p2 l 8-10 – and vegetation WUE / transpiring properties

p2 l 17 If you talk about climate, do you mean balance between evaporation and precipitation? Transpiration is not purely climatic.

Section 2.1 – Just checking, is there any permafrost in northern catchments, and if so, can there be any impact (e.g. thawing permafrost -> higher storage)?

p4 l 1 – how it was calculated?

Formula 2 – why in the middle line Pi = 1? What does 1 mean?

Section 4 – Can the changes in Sr be related to changes in WUE (e.g. Troch et al 2009)?

p9 l 8-9 – Is it just direct numerical effect of having higher runoff from drained peatlands?

p9 l 11 – suggest changing 'many affects to' to 'many effects on'

p9 l 15-16 – I still struggle with the idea of how Sr calculated with pan evaporation would change if only vegetation properties change (as 'or' implies independency) – see my general comment in the beginning. In any case, the argument is based on the assumption of trading space for time (Wagener et al, 2007, Singh et al, 2011), and this and associated assumptions can be acknowledged better (possibly also in introduction).

References:

Saft, M., Peel, M. C., Western, A. W., & Zhang, L. (2016). Predicting shifts in rainfall-runoff partitioning during multiyear drought: Roles of dry period and catchment characteristics. Water Resources Research, 52(12), 9290-9305.

Singh, R., Wagener, T., Van Werkhoven, K., Mann, M. E., & Crane, R. (2011). A trading-space-for-time approach to probabilistic continuous streamflow predictions in a changing climate-accounting for changing watershed behavior. Hydrology and Earth System Sciences, 15(11), 3591.

Troch, P. A., Martinez, G. F., Pauwels, V. R., Durcik, M. , Sivapalan, M. , Harman, C. , Brooks, P. D., Gupta, H. and Huxman, T. (2009), Climate and vegetation water use efficiency at catchment scales. Hydrological Processes., 23: 2409-2414. doi:10.1002/hyp.7358

Wagener, T. (2007). Can we model the hydrological impacts of environmental change?. Hydrological Processes, 21(23), 3233-3236.

---

## Author Comment (AC3) · 20 Jun 2018

Dear referee,
Thank you for reviewing our manuscript, your positive evaluation of the relevance and presentation of the results and the relevant questions you posed. The more detailed comments you have given made us realise that the aim we have in mind with the paper needs to be discussed better throughout the manuscript and that especially some elements of the used method need more attention in a revised version of the manuscript. Below we have replied in more detail to all your comments.

*The study is on an important and interesting topic, as little is known about potential changes in catchment storage properties under climatic and land use change. Potential*

[Figure]

*to improve our hydrological predictions under climatic and land use change is limited by lack of information and understanding, so studies in this area can be expected to be in demand by HESS audience. Another strength of this study is the dataset, which is very well described and referenced. The article is nicely structured, and the results are presented well (though I would prefer to see more numerical information to back up some claims).*

*My main concern about this study is about how much of the results originate from self-prediction given the high correlation between source and comparison data. $S_r$ is derived based on climatic records, and then $S_r$ is compared with climatic and vegetation properties which are known to be related with the source data $S_r$ was derived from. The authors do acknowledge the relationship (e.g. p 5 l 24-25, p 6, l 2-3, p 9 l 16-17), but they still interpret results in a way where (higher) correlation implies control over $S_r$, which I think is questionable. The results might reflect just the closer correlation with the source climatic data for $S_r$, and not causal relationship with the soil storage properties. For example, it remains unclear to what extent the relationship between $S_r$ and vegetation properties/land use are just a consequence of both being related to the climate. In this case the change in vegetation would not influence $S_r$ as it can be expected from the results, if the vegetation is the only thing changing. This would particularly apply to cases where the results are somewhat counterintuitive (e.g. p6 l26-27 and Fig 4a, or p7 l23-25 and Fig 7b where decrease in forested area is associated with increase in $S_r$).*

We agree with you that the variables used in the analysis are subject to internal correlations. As discussed in the reply to the review of Maik Renner, the exact variables used for the calculation of $S_r$ are not the same as the ones used in the remainder of the analysis.

With respect to the influence of vegetation on $S_r$, this can be reflected by using a different drought return period (Wang-Erlandsson et al., 2016): different vegetation types probably have different survival strategies and therefore are likely to adjust to a

different drought return period. However, for this study we only used a 20-year return period as the majority of the land cover consists of forest. So, if only the vegetation would change, a different return period can be used, which would influence the derived $S_r$. However, more testing would be required to see if changes can be assessed by using different return periods and how (quickly) new equilibriums would be established.

In addition to this, the catchments with a lower forest cover are generally the ones with a higher agricultural cover and a milder climate. These catchments are likely to have higher transpiration demands, leading to higher $S_r$ values, than colder forested catchments.

Having said this, the term 'control' might be misleading, especially as our main aim with the study is to compare the calculated $S_r$ values with a set of catchment characteristics to explore possible relations and better understand the derived $S_r$ values and what influences their calculation. Your comments and those of the other reviewers made us realise that we did not discuss this aim consistently throughout the paper and in the revised version of the manuscript we will make this clearer.

*In this light, I think it would be more informative and would give more confidence in the results to apply some method which can account for a number of potential "controls" and assess their importance against each other, for example PCA or multimodel inference (e.g. Saft et al, 2016).*

Thank you for this suggestion; we think as well that the paper would benefit from such further elaboration. Both multimodel inference (as used by Saft et al., 2016) and PCA are useful methods for this. In line with our methodology, we will work this out further using the PCA. However, we do believe that the multimodel inference is very interesting when evaluating the possibilities to use climate derived $S_r$ values to assess the effects of change.

The results of a PCA are presented in Figure 1, containing the different variables we

compared with $S_r$ in the manuscript. As already expected, this analysis shows that the majority of the climate variables (shown in blue) are positively correlated and negatively correlated to the mean annual temperature and transpiration demand (which follows from the water balance). What can also be seen is the limited correlation between the majority of the climate variables and (summer) precipitation.

With respect to vegetation properties (shown in green), these are strongly correlated with forest and agricultural land covers, but limitedly correlated to the majority of the climate variables. Only peatland covers are positively correlated with the majority of the climate variables.

The plotted catchments (top plot) indicate that the eco-regions mainly differ in climate characteristics and that especially in the mid- and south boreal regions a large range of vegetation properties and land covers occur.

We will incorporate these results in a revised version of the manuscript.

*I am also a bit puzzled about the gap between snowmelt and onset of PET, as both are governed by exactly the same increasing energy flux (temperature/sunshine). I would assume that this gap should be very closely related to the maximum SWE (âĹijmore snow takes longer to melt). Anyway, it would be interesting to calculate this gap (using some threshold for snowmelt) and include it directly as yet another factor along with the other characteristics used. I wonder why it was treated separately.*

$E_p$ and snowmelt are indeed governed by the same increasing energy flux. However, the gap exists because of the measurement methods: $E_p$ is based on pan-evaporation and can thus only be measured if temperatures are above zero. Therefore, $E_p$ can already be slightly above zero before the pan measurements start.

With respect to treating both variables separately, this was done because in the $S_r$ calculations one determines the water demand, the other the water supply. The balance between these variables mainly determines the calculated $S_r$. So, although they are

governed by the same energy flux, they have a different effect on the calculation. For the aim of the paper, we think this influence is interesting to investigate and explore.

Having more insight into the combined effect of the onset of $E_p$ and snowmelt can help to assess what can happen in case of a changing climate. For the gap between these two to change, the most important variable will indeed probably be the maximum $S_{SWE}$.

The gap between snow-off and the start of $E_p$ (measurements) is incorporated in the PCA presented in Figure 1. It can be seen this property is strongly positively correlated with the day and amount of maximum $S_{SWE}$.

*On a different note, it would be good to see more numerical information (i.e. Spearman's rho, and associate p value) associated with positive/negative correlations described in the text. It is difficult to extract relevant information from figure 8, especially since it is not numeric. Fig 8 also does not include correlation results for sub-regions which are mentioned in the text, and I could not find these results anywhere else.*

As discussed in the replies to the two other referees as well, we will include more numerical information about the correlations in the text, both for the entire data set as for the sub regions. We preferred not to include them in Figure 8, to prevent it from overflowing with information. However, for the revised manuscript we will see if there is a way to include some more numerical information in the figure as well.

**Importance and implications:**
*What is the use of the derived $S_r$ and discovered relationships with other characteristics? And in the context of climate change, would not it be easier to derive new $S_r$ following the original method accounting for climate change in the source data instead of looking at the correlations?*

As discussed before, our main of the paper is to explore the relations between the

climate derived $S_r$ and a set of catchment variables. We should keep in mind that $S_r$ is a conceptual parameter, originally used as input for hydrological models. However, it is very interesting to know if and how the calculated $S_r$ is related to other variables and if it can be wider applicable. Knowing more about the relations between $S_r$ and catchment variables can help us to better understand the influences on the $S_r$ calculations and therefore how we can, possibly, use it to assess the behaviour of catchment under changing conditions.

Thus, by looking at the relation between $S_r$ and catchment variables, more confidence can be obtained in the meaning of $S_r$. The found relations are not directly meant to assist to assess change. To assess change indeed recalculating $S_r$ based on new climate predictions would be the most logical approach.

We will make the division between better understanding $S_r$ and using $S_r$ for assessing effects of change clearer in the revised version of the manuscript.

**Specific comments:**
*p2 l 8-10 – and vegetation WUE / transpiring properties*

The partitioning is indeed influenced by both water availability (in the root zone storage) and water use efficiency of the vegetation. However, with the vegetation surviving in a certain catchment, it must have had sufficient water supplies and at the same time it is not logical that it would have invested more carbon in creating storage capacity for water than it needed to survive. So, by deriving the transpiration demand from the water balance, the long term water use is estimated, making the water efficiency of the vegetation less relevant for the calculation. The balance between transpiration and runoff is of course influenced by the water use efficiency of plants.

We will acknowledge this aspect in the revised version of the manuscript, together with a more extensive description of the calculation method for $S_r$ and the assumptions involved.

*p2 l 17 If you talk about climate, do you mean balance between evaporation and pre-cipitation? Transpiration is not purely climatic.*

Actually we are talking about the difference between the long term average supply and demand of water to the active storage of the soil. These are mainly climatically driven (ie. via precipitation and potential evaporation). As processes like interception and snow melt are important as well, we decided to use the terms infiltration and transpi-ration (demand). Where infiltration is the total precipitation (rainfall and snow) minus interception evaporation and the transpiration (demand) is derived from the water bal-ance ($\overline{T} = \overline{P} - \overline{E_i} - \overline{Q}$). We will clarify these terms in a revised version of the manuscript, mainly by extending the description of the method used to calculate $S_r$.

*Section 2.1 – Just checking, is there any permafrost in northern catchments, and if so, can there be any impact (e.g. thawing permafrost → higher storage)?*

Thanks for asking this clarification, but in these sites there is no permafrost. If there would be, changes in permafrost should indeed be included in the calculation of $S_r$, just like snow storage is.

*p4 l 1 – how it was calculated?*

Data of all three biomass variables (root biomass, tree height and leaf cover) are based on field data from national forest inventories, satellite images, digital map data and other georeferenced data sets. More information can be found in Mäkisara et al. 2016 (http://jukuri.luke.fi/handle/10024/532147).

*Formula 2 – why in the middle line Pi = 1? What does 1 mean?*

Thank you very much pointing this out: it should be Pi = 0. We will correct this in the revised version of the manuscript.

[Figure]

*Section 4 – Can the changes in $S_r$ be related to changes in WUE (e.g. Troch et al 2009)?*

In this study we did not yet incorporate any changes in the catchments, so we suppose you mean the variations in $S_r$ between the different catchments. The calculation of $S_r$ is based on a daily simulation of soil moisture deficit and an extreme value distribution. The input into the simulation is effective precipitation and a transpiration demand which is estimated from the water balance. As the transpiration demand is the water that should have transpired to close the water balance, different water use efficiency probably will not really influence the derived $S_r$ values. However, water use efficiency could influence the amount of biomass production (root biomass, leaf cover, tree height). Thank you for this suggestion; we will elaborate this in more detail in a revised version of the manuscript.

*p9 l 8-9 – Is it just direct numerical effect of having higher runoff from drained peatlands?*

One of the assumptions underlying the climate derived root zone storage capacity is that a certain type of vegetation needs a specific amount of water to survive; independent of the climate they are located. If they are located in a drier or more seasonal climate, they will need a larger storage capacity to supply the required amount of water. This does neglect the fact that vegetation may have higher water use efficiency (Troch et al., 2009). As discussed earlier, this does not influence the calculation, as the used transpiration demand is derived from the water balance. By deriving the transpiration demand from the water balance, the runoff is already accounted for. However, differences in water use efficiency could help to explain the pattern found for the pine root biomass. We will elaborate this in more detail in a revised version of the manuscript.

*p9 l 11 – suggest changing 'many affects to' to 'many effects on'*

We will change this in the revised version of the manuscript.

*p9 l 15-16 – I still struggle with the idea of how $S_r$ calculated with pan evaporation would change if only vegetation properties change (as 'or' implies independency) – see my general comment in the beginning. In any case, the argument is based on the assumption of trading space for time (Wagener et al, 2007, Singh et al, 2011), and this and associated assumptions can be acknowledged better (possibly also in introduction).*

$S_r$ is based on a daily simulation of soil moisture deficit and a drought return period the vegetation adapts to. Different vegetation types are likely to adjust to different return periods, as they have different survival strategies. In addition to that, $E_p$ data is only used to add seasonality to the long term averaged transpiration demand, which is derived from the water balance (precipitation and runoff). So, a change in vegetation probably works in two ways: a different drought return period can be applicable and of course the balance between precipitation and runoff can change. In either way, a new equilibrium needs to be established.

In addition, the principle of trading space for time can be used as well, especially as we see shifting conditions in the study areas. However, to go into that direction in this paper would make it lose focus, but we will make the change in vegetation properties we had in mind clearer. Although we have discussed the change part in the paper, and we definitely think a thorough understanding of a climate derived $S_r$ can help to assess hydrological change, our results do not discuss any of these elements.

Having said this, in a revised manuscript we will make clear that the aim of the paper is to better understand the climate derived $S_r$ and its relation with certain catchment properties and mention the change as an outlook and an, in our opinion, important possible applicability.

**References:**

Saft, M., Peel, M. C., Western, A. W., & Zhang, L. (2016). Predicting shifts in rainfall-runoff partitioning during multiyear drought: Roles of dry period and catchment char-

**HESSD**

Interactive
comment

acteristics. Water Resources Research, 52(12), 9290-9305.

Singh, R., Wagener, T., Van Werkhoven, K., Mann, M. E., & Crane, R. (2011). A trading-space-for-time approach to probabilistic continuous streamflow predictions in a changing climate-accounting for changing watershed behavior. Hydrology and Earth System Sciences, 15(11), 3591.

Troch, P. A., Martinez, G. F., Pauwels, V. R., Durcik, M. , Sivapalan, M. , Harman, C. , Brooks, P. D., Gupta, H. and Huxman, T. (2009), Climate and vegetation water use efficiency at catchment scales. Hydrological Processes., 23: 2409-2414. doi:10.1002/hyp.7358

Wagener, T. (2007). Can we model the hydrological impacts of environmental change? Hydrological Processes, 21(23), 3233-3236.

Wang-Erlandsson, L., Bastiaanssen, W. G. M., Gao, H., Jägermeyr, J., Senay, G. B., van Dijk, A. I. J. M., Guerschman, J. P., Keys, P. W., Gordon, L. J., and Savenije, H. H. G.: Global root zone storage capacity from satellite-based evaporation, Hydrol. Earth Syst. Sci., 20, 1459-1481, https://doi.org/10.5194/hess-20-1459-2016, 2016.

————————————————————

**Fig. 1.** Results of principal component analysis with variables compared to Sr. Top plot shows catchments in the three boreal regions

---

## Author Response (AR1)

**Author's response for review process of "Understanding variability in root zone storage capacity in boreal regions"**

by Tanja de Boer-Euser, Leo-Juhani Meriö, Hannu Marttila

Dear editor,

Please find attached the revised version of our manuscript. The comments of the reviewers have been very useful for this revised version. This document first contains a point-to-point reply to all their comments, the core of this point-to-point reply is the same as what we posted in the interactive discussion, with details included how comments were finally incorporated. The point-to-point reply is followed by a marked-up version of the revised manuscript and supplement.

The most important changes in the manuscript are:

- We have made the aim of the paper clearer by rewriting, among others, the last paragraph of the introduction and the first paragraph of the discussion.

- We have changed the structure of the paper, to make the description of the method, the presentation of the results and discussion of the results more consistent. In the revised version the Sections 'Methods', 'Results' and 'Discussion' have, among some other subsections, the subsections: 'Climate variables', 'Vegetation characteristics' and 'Vegetation types'.

- We have extended the description of the method to derive $S_r$ from climate data.

- We have added a principal component analysis (Sections 2.4.4 and 3.1) and as a consequence we have moved the correlation matrix (Figure 8 in the first version of the manuscript) to the supplement, to prevent overlapping information.

- We have added the Spearman's correlation coefficients in the titles of the subplots of the scatter plots comparing $S_r$ with climate variables, root biomass and vegetation types, for all catchments combined and for the individual boreal regions. The Spearman's correlation coefficients for the comparison of $S_r$ with leaf cover and tree height are mentioned in the text.

- Due to the change in structure of the method, result and discussion sections, the order of the figures changed in the following way:

| revised version | first version |
| --- | --- |
| Figure 1 | Figure 1 |
| Figure 2 | Figure 2 |
| Figure 3 | not yet present |
| Figure 4 | Figure 5 |
| Figure 5 | Figure 6 |
| Figure 6 | Figure 4 |
| Figure 7 | Figure 3 |
| Figure 8 | Figure 7 |
| Figure 9 | Figure 9 |

We would like to submit this revised version for the next step in the review process.

On behalf of all authors,

Kind regards,

Tanja de Boer-Euser

**1    Reply to review of anonymous referee #1**

Dear referee,

Thank you for reviewing our manuscript and your positive evaluation of the general findings. The more detailed comments you have given made us realise that especially some elements of the used method need more attention in a revised version of the manuscript. Below we have replied in more detail to all your comments.

*A weakness in the analysis is that $S_r$ is derived at the basin scale, and then assessed against vegetation type and attributes, but different vegetation types prefer different soil texture and moisture conditions. A diverse catchment likely has a diverse soil and an $S_r$ value that may not apply very well to any of the vegetation types in the particular catchment. Conversely, I would expect stronger correlations and more valid $S_r$ values in catchments containing a dominant vegetation type. From looking at Figure 3, I suspect that peatlands never make up enough of the catchments for the $S_r$ value obtained to be applicable to them, and this would also be the case for many of the agricultural areas. Forest is usually the dominant vegetation type and this is shown by its close agreement with the broad boreal zone plot.*

We agree with the reviewer that stronger correlation between $S_r$ and vegetation characteristics could have been found if the catchments contain only one dominant vegetation type. We used existing small catchment data which allowed us to compare catchment attributes to $S_r$-values, but boreal catchments are rather heterogenic and thus suitable data from catchments with single vegetation type does not exist. However, the calculation method assumes equilibrium in the catchments: all vegetation in the catchment managed to survive there together. This means that the catchment representative root zone storage capacity is appropriate to sustain the transpiration demands of the vegetation. Although $S_r$ is a catchment average conceptual parameter we expected that is related to different vegetation characteristics, which we have shown in the results.

Regarding the land cover types in Figure 3, we agree with the reviewer that the derived $S_r$ values cannot fully be attributed to one of the land cover types; however, by presenting the results in this way, we think that possible influences of certain land cover types on the derived $S_r$-values can be explored.

We have discussed the effect of the heterogeneity of the catchments in Section 4.2 of the revised version of the manuscript.

*There appear to be inconsistencies in the data that are presented in subsequent plots. These need to be corrected or explained. In Figure 3a there are two points with relatively low leaf cover and $S_r$ values of about 230 and 110 mm. These points show up in the boreal regions plot (3d) as northern points, so they are northern forests. Based on Figures 3e and 3h, the 230 mm $S_r$ value is also associated with a large tree length of about 210 m while the 110 mm $S_r$ value is associated with a medium tree length of about 100 m. I'm wondering if there is something about these two basins that makes them different. Why is the leaf cover low and yet one of them has the largest observed tree length? Was there a defoliation event? When I look in Figure 4, I see no northern forests with an $S_r$ value anywhere near 230 mm. At first I thought that perhaps it wasn't pine, spruce or deciduous, but it doesn't even appear in figure 4d. What happened to this forest that stands out in Figure 3? Figure 3 shows two northern catchments with $S_r$ values not far from 110 mm, one is mostly forest with some peatland and the other has a bit less forest but still more forest than peatland. Figure 4 only shows*

*one northern catchment with $S_r$ values close to 110 mm. What happened to the other catchment? Figure 3 shows two northern catchments with $S_r$ of 70-80 mm and one at about 50 mm, and these appear to be forests or mostly forests, but in Figure 4 the $S_r$ values do not fit the same distribution of two in the 70-80 mm range and one at 50 mm but instead it appears that two are at about 70 mm and one at about 85 mm. These plots appear to be derived from somewhat different datasets with respect to $S_r$ values. The data used in the figures needs to be made internally consistent, or explanations provided for data appearing in some figures and not in others*

Thank you very much for pointing this out. In Figure 3 the data for leaf cover was slightly shifted, creating an inconsistency in the presented data. We have corrected this error, resulting in a new figure, being Figure 7 in the revised manuscript. Further, the x-axis caption is changed from tree length to tree height for more clarity and data were presented in decimeters, which is now converted to meters.

*From my experience, some pine species like to grow in sandy well-drained soil, and here contribution to discharge is likely high and transpiration low. In such a catchment the estimated T should be low and there will not likely be large deficits, even though the soil can get quite dry. Spruce trees like to grow in moist soil, often in poorly drained areas. Such areas don't often dry out and contributions to discharge also likely follow precipitation quite well, except following a drought when there is recharge; again such areas may not see very large deficits. So we have pine in dry areas with small deficits and spruce in wet areas with usually small deficits.*

We agree with the reviewer. Pine trees favour dry sandy soils whereas spruce favour more moist locations. However, in our data set there were also many pine trees in drained peatlands and thus we cannot fully follow this simplification in our analysis. The small expected deficits for both pine and spruce trees, is also reflected in the derived $S_r$-values: many catchments have $S_r$ values below 100mm and spruce and pine are the dominant tree species. With the used method $S_r$ values larger than 500 mm are found worldwide (e.g. Gao et al., 2014; Wang-Erlandsson et al., 2016). We have discussed the possible influence of the drainage capacity of the soil in Section 4.3 of the revised manuscript.

*Deciduous trees tend to have larger transpiration demands and can grow in poorly or well drained soils. If deciduous trees exist more often in areas with larger deficits and adjust their root mass accordingly, this may explain why the best correlation is for deciduous trees in Figure 4.*

Thank you for pointing this out, we have briefly discussed this aspect in Section 4.3 of the revised manuscript.

*However, much of this detail would be smeared out because each basin contains multiple tree and other vegetation types and probably a combination of wet and dry areas. With this in mind, I understand why the correlations and patterns are not as strong as one might hope for.*

As mentioned before, the study catchments were indeed rather heterogenic as typical boreal landscape is. However, some correlations were found between derived $S_r$-values and detailed catchment data. We have discussed the heterogeneity of the catchments in Sections 4.2 and 4.3 of the revised version of the manuscript.

*Some of the relationships appear to be curvilinear rather than linear, so it might be more informative to try fitting some nonlinear relationships (exp, log, polynomial) to see which correlations increase and whether the*

*relative importance of parameters changes. Perhaps a flexible generic nonlinear model could be used.*

Thank you for the good suggestion; however we do not see need for the non-linear methods since results can be shown in linear methods. Actually, we think it is more valuable to combine the linear methods with an analysis of the possible threshold present in the data. This threshold seems to turn up at the same location for a number of the compared catchment characteristics. Significant differences between 'below' and 'above' threshold groups are presented in Section 3.5 of the revised manuscript.

**More specific comments:**

*P4 line 4-7: Are the authors aware of the type of precipitation gauges used to measure snowfall, whether they were shielded and whether they were corrected for undercatch based on coincident wind speed measurements? Precipitation gauges always measure less than the true snowfall amount, but if properly located, shielded and adjusted using established correction factors based on wind speed, one can arrive at an accuracy that is comparable with a snow survey.*

We used a spatially interpolated dataset with a resolution of 10 x 10 km$^2$ for the meteorological parameters (precipitation, air temperature) constructed by Finnish meteorological institute (FMI). In this data set the measurement error caused by gauges has been checked and corrected in operative quality control. For snow data ($S_{SWE}$), we used snow line data provided by Finnish Environment Institute and measured by standard methods. Since $S_{SWE}$ was closest available and not always situated within the study catchment, we corrected $S_{SWE}$ with local precipitation. We have added the following sentence to Section 2.2: 'These data have been checked for measurement errors caused by gauges and were corrected in operative quality control.'

*P4 line 22-23: I am somewhat perplexed that canopy interception is included for rain but not for snow, when it is well known that boreal forests can store close to an order of magnitude more mass of snow versus water on the canopy, and interception losses on the order of 30% or more are common over a winter.*

We used snow line data to provide snow water equivalent values for the $S_r$ calculations. This data represent rather well the snow water contributing to the runoff, soil moisture and recharge, and is therefore suitable for our analysis. In addition to that, the interception included in the calculations to estimate $S_r$ is, besides the availability of water, driven by the potential evaporation. As the latter is not measured, as it is close to zero, during winter time (so during occurrence of snow), including interception for snow would not really influence the estimated $S_r$ values.

*P5 Section 2.4: I think an explanation of the specific method used to obtain $S_r$ is required. I looked at de Boer-Euser et al. (2016) and based on that, I think I understand what was done, but a brief overview would be helpful.*

We have extended the explanation of the method in Section 2.3 of the revised manuscript, including assumptions of the method and terminology used in the remainder of the manuscript.

*In Figure 3 are the values of leaf cover and tree length basin values or are they specific to each vegetation type? I see for example the two northern basins with $S_r$ near 70-80 mm and leaf cover near 24-28% in Figure 3d, and these appear to have corresponding large forest fractions, small peatland and smaller agricultural fractions with the same leaf cover values. This suggests that these values are basin-scale and are not specific to each*

*vegetation type. Since most of the basins are forest-dominated, when we look at the peatlands or agricultural plots, in most cases when the fractions of these vegetation types are small, we are not looking at leaf cover or tree length values that have anything to do with the peatland vegetation or crops other than they happen to be in the same basin. This should be made more clear.*

Values in Figure 3 are basin scale and thus not specific for the vegetation types. Figure 3 illustrates the general patterns of $S_r$ value in boreal catchments and variation with main landscape types. It cannot be used to detect vegetation type changes. We have changed the text presenting the results in this figure (Section 3.3 in the revised manuscript). The figure is now introduced with the following sentence: 'For both comparisons the data is plotted indicating the occurence of different vegetation types (forest, pristine peatlands and agriculture) in the catchments and the boreal regions in which the catchments are located'

*P6 line 19: The statement "...and this correlation decreases for higher percentages of peatland..." is a bit misleading. There hasn't really been an analysis of correlation for basins with high and low peatland cover. When I look at Figure 3f, it does appear that there may be some correlation between $S_r$ and tree length in pristine peatlands for the basins with small fractions of pristine peatlands (because the correlation is coming from the larger forest fractions) and the pattern looks more scattered (implying a lower correlation) for the larger circles or basins with a larger peatland fraction. It should be made clear that these are just visual interpretations, not a comparison of calculated correlation coefficients.*

We agree that this statement is rather strong. We have changed the sentence into: 'When looking at the different vegetation types, it can be seen that catchments with a large forest cover are the ones with the widest range in leaf cover and tree length.' in the revised version of the manuscript. Further, we have added more numbers in the text and specifically mentioned if results were only visually observed from the graphs.

*P6 line 20: The variability in leaf cover and tree length is small within the boreal regions but appears greater when the three regions are examined together. It appears that factors affecting tree length and leaf cover act largely but not exclusively along the latitudinal gradient such that the correlation is weak within each region. I think the strong relationship between Day of Year (date of snow-off) and $S_r$ has more to do with the fact that the snow-off date is correlated with both maximum SWE and air temperature than a special relationship with the phase of snowmelt. For example, the timing of maximum SWE is probably determined almost exclusively by temperature, whereas the amount of maximum SWE is a combination of snowfall amount and temperature (and other factors).*

We agree with you: latitudinal and climate gradient in data set affects strongly to the results, which we shortly discussed in Section 4. We have discussed this in more detail in Section 4.1 of the revised manuscript. For this point please also refer to our reply to the comment about the principal component analysis of anonymous referee #2.

Regarding the influence of snow-off date on $S_r$, we agree there is a strong link between max $S_{SWE}$, mean air temperature and snow-off date. However, from our analysis it turned out that especially the timing of the snow melt is important. Although this is strongly determined by temperature, it is not directly reflected in the mean annual air temperature. Various studies using the climate derived $S_r$ showed that mainly two variables are important for $S_r$: the absolute difference between water supply (liquid precipitation or snow melt) and water

demand (transpiration) and the phase difference between these two (ie. difference in timing of the majority of the supply and demand). In areas with moisture constrained evaporation the absolute difference is likely to be dominant, while in energy constraint (like boreal areas) the phase difference is likely to be dominant. So, the study areas have similar absolute differences between supply and demand on a yearly basis, while the phase difference strongly differs depending on the snow-off date and onset of potential evaporation. We have discussed this aspect in Sections 2.4.1. and 4.1 of the revised manuscript.

*P7 line 22: While it is true that the clearing of land for agriculture increases soil exposure (more evaporation) and crops tend to have high transpiration demands (more transpiration), there is also the likelihood that croplands are more prevalent in the south because of the longer growing season and increased likelihood of a successful crop. So did the crops in the south cause larger $S_r$ values because of their higher water demands or were they planted in a warmer area because it is beneficial for the crops and that just happens to coincide with larger $S_r$ values (warmer, more evaporative demand)? I would say it works in both directions.*

Thank you for pointing this out; we agree that this works in both directions. To investigate whether one of the two mechanisms is dominant, a more detailed comparison should be made between catchments in the southern region with more and less agricultural cover or between different periods of the same catchment (before and after clearing). We have discussed this in Section 4.3 of the revised manuscript.

*P8 lines 1-3: Peatlands generally develop in areas where the soil does not dry out very often, either because of cold temperatures and low evaporative demand, or a combination of positive P-E and poor drainage. Since the soil does not tend to dry out, the $S_r$ value calculated will be small because large deficits of P-T are rare.*

We agree with you that $S_r$ values will be small due to small deficits of $P - T$. We have discussed some other reasons for low $S_r$ values in peatland areas in Section 4.3 as well.

*P8 line 11-12: Maximum SWE and mean annual temperature and the snow-off date are likely highly correlated within a small region. A regression model that attempted to include all three would almost certainly show that all three are not necessary. I would be inclined to predict that mean annual temperature and maximum SWE are the most important, but maximum SWE is partially dependent on mean annual temperature based on the length of the snow period and when melt starts. Perhaps mean annual temperature and winter precipitation would do better.*

In Finland and in our data set there is latitudinal and longitudinal variation in maximum $S_{SWE}$, mean annual temperature and snow-off date since different areas are affected by either Atlantic (Western areas), Continental (Eastern areas) or Arctic (Nordic areas) weather patterns. Thus, all of these parameters are relevant to include into the analysis, although correlation between them exists (see also our reply to the comment of anonymous referee #2 about a principal component analysis of the tested variables). And as discussed before, the timing of the water supply is very important for $S_r$, thus so is the snow-off date.

*P9 line 6: I have read that jack pine have a tap root to access deeper water. If this is true of the pines in the Finland catchments, it may be that deeper water is accessed without a large increase in root density and this may lessen correlations between $S_r$ and root biomass. The authors would want to find an appropriate citation before using this point as an explanation.*

Variability in S$_r$ in boreal regions - author's response 7

Scots pines have shallow rooting depth and typically have no access to deeper water. Also in our data set many mid- and south boreal catchments contained drained peatlands which might influence this negative correlation.

*P9 lines 1-3: Yes, peatlands develop in places where the decomposition rates are slower than the annual increment, due to a combination of cold temperatures and/or poor drainage and anoxic conditions. Peatlands are created by the same conditions that cause the estimated $S_r$ to be low, but I doubt that peatlands cause the small $S_r$ values.*

The climate method uses the assumption that equilibrium exists in the catchments between the existing vegetation and the develop root zone storage capacity. As the root zone storage capacity is a catchment representative value, it is 'caused' or created by the combination of all the vegetation (and thus land cover) in a catchment. So, we agree that the peatlands alone do not cause a low $S_r$ value, but they probably contribute to it. We have clarified this in the revised version of the manuscript by extending the description of the method to derive $S_r$ and by adding a part in the discussion about the effect of the heterogeneity of the catchments on the derived $S_r$ values. Further we have focused more on relations between $S_r$ and other variables and less on a causal relation between the two, as discussed in the replies to the review of anonymous referee #2 as well.

**Minor comments and corrections:**

*$S_r$,20 is never defined in the text. It is stated in Section 2.3 that a drought return period of 20 years is used, but the symbol $S_r$,20 is not introduced here; it simply appears in figures but not in the text.*

We have changed '$S_{r,20}$' to '$S_r$' in the figures.

*P4 line 21 and elsewhere: Why is $S_{SWE}$ used for Snow Water Equivalent instead of SWE?*

Although SWE is more common in literature, we used $S_{SWE}$ (Storage as snow water equivalent) to prevent abbreviations with multiple capital characters, as is requested in the author guidelines of the journal.

*P5 line 14: "Tree length" is never defined. It is certainly not tree height, but I don't see the term in the literature.*

The variable presented in figure 3 is actually tree height, but in dm and not in m. We have changed the term to tree height and the data in Figure 3 (Figure 7 in the revised manuscript) is now presented in meters.

*P7 line 24-25: In Fig. 7c I might view the southern boreal region as showing a negative correlation between Drained peatland % and $S_r$ with two outliers.*

Thank you for pointing this out, we have changed the sentence as follows: 'The drained peatlands (Figure 8c) also show a negative correlation with $S_r$ when considering all catchments and for the mid-boreal region: for the north and south boreal regions no significant correlations were found'

*P9 line 24: I would change "…. for example indicates that…." to "…for example may indicate that…."*

We have changed the text accordingly.

*Figure 1: Add a North Arrow. Perhaps outline Finland so as to make the study area boundaries more clear.*

We have added a north arrow to the map and the country boundaries are presented.

*Figure 3: The letters need to be on the plots (e.g. a, b, c.... h).*

The letters are added to the subplots (Figure 7 in the revised manuscript)

*Figure 6: Change Julian date to Day of Year. Julian date or Julian day is not the same as Day of Year.*

The label on the axes is changed to day of the year

*Figure 8: What do the size of the boxes represent? There is no scale provided to interpret this.*

The sizes of the boxes indicate the p values of the correlations; we have added this to the caption. Note though that we have moved this figure to the supplement, as it has a lot of overlapping info with the correlation coefficients now presented with the scatterplots and the principal component analysis (Figure 3 in the revised manuscript).

**2  Reply to review of Maik Renner**

Dear Maik Renner,

Thank you for the review of our manuscript and the interest in our study. Your comments made us realise that some elements are not yet explained well and that our argumentation misses some intermediate steps. Therefore they were very valuable for improving our manuscript. Below we have replied in more detail to all your comments.

*The results show that S increases towards the south, increases with biomass, but decreases with area of peatlands due to high water tables. Since root biomass is used as a metric for verification, more details are required on how this was derived.*

The root biomass data is based on multi-source national forest inventory data provided by Finnish Natural Resources Institute (LUKE). The data is based on field data, satellite images, digital map data and other georeferenced data sets. More information can be found from Mäkisara et al. 2016. `http://jukuri.luke.fi/handle/10024/532147`. We have added this information in Section 2.2.

*The statistical analysis is presented in a way to suggest that root zone storage is independent of the climate variables ($P$, $E_p$, $T$, $SWE$), while indeed it is derived from these data. Actually the analysis of climate controls is performed like an uncontrolled sensitivity analysis of a bucket model with different inputs. The outcomes of this sensitivity analysis (Fig 6, Sect. 3.2, 4.2) are difficult to interpret since the influence of the other input parameters changes from one catchment to the next. I also wonder why there is no precipitation frequency / drought index be used to correlate with S?*

We agree with you that the root zone storage capacities are dependent on climate variables. Actually, for the calculation of $S_r$ four climate parameters are used, namely: daily precipitation, daily snow water equivalent, long term averaged discharge and long term monthly averaged potential evaporation. Although climate parameters were used in Figure 6, these are not variables that are directly used in the calculations (mean annual temperature, maximum snow water equivalent, snow off date and the ratio of precipitation and potential evaporation).

As the estimation of $S_r$ is not one calculation, but derived from the simulated soil moisture deficit (we have made this clearer in the description of the method) the influence of different climate variables is not always straight forward. Therefore, we used these plots to see if there is any correspondence between $S_r$ and climate variables that were not directly used in the estimation of $S_r$. We realise however, that we made this aim not fully clear in the discussion of the results and we have changed the text accordingly in the revised version of the manuscript. In our reply to the comment of anonymous referee #2 about a principal component analysis more details about correlations between variables are described.

With regard to the precipitation frequency/drought index: we included $P/E_p$ in the analysis, which is a definition of the aridity index. Further, we compared $S_r$ with runoff coefficients during the analysis, which showed a strong relation. This is logical, as it is one of the main inputs in the estimation of $S_r$ and probably even has a stronger effect on the calculation than $S_{SWE}$; therefore we considered the other plots more interesting to incorporate in the manuscript.

[Figure]

**Figure 1:** Average interstorm duration in relation to derived root zone storage capacities, different symbols indicate different boreal regions (green squares = south boreal; blue circles = mid boreal; red triangles = north boreal)

With respect to precipitation frequency a comparison with inter-storm duration ($I_{isd}$) could be made; we did not do this during the first analysis. The relation between $S_r$ and $I_{isd}$, based on total precipitation and an interception capacity of 1.5 mm, can be seen in Figure 1 below. It can be seen that the variability in $I_{isd}$ between the catchments is very limited. Therefore, we do not think that adding this plot in the revised manuscript is valuable for the analysis.

*An interesting point is the influence of drainage of peatlands on S. Although the authors claim to identify an effect, I could not identify the mentioned influence of drainage in Fig 7. Unfortunately, the analysis lacks a reference to compare drainage with pristine peatlands. Here a stratification of the data could be useful means to assess this point.*

From Figure 7 (Figure 8 in the revised manuscript) it can be seen that more catchments exist with larger $S_r$ values and larger percentages covered with drained peatland than with larger $S_r$ values and larger percentages covered with pristine peatlands. The corresponding correlation coefficients show a negative correlation between $S_r$ and both pristine and drained peatlands; however this correlation is stronger for the pristine peatlands. Two reasons for this difference can be given. First, the drainage of peatlands for forestry probably creates larger transpiration demands and thus larger root zone storage capacities. Or, second, as most of these drained catchments are located in the south boreal region, it can also be that $S_r$ values were already higher in these catchments before the peatlands were being drained. Unfortunately the available data series are not long enough to compare $S_r$ values before and after drainage of some of the drained catchments. We have discussed this in Section 4.3 in the revised manuscript.

*I do not understand how the method can be applied in climate or land-use change analysis. To my understanding an estimate of transpiration is required to estimate S and both are unknown for a given change scenario. Please explain.*

For climate and land-use change analysis often data are available for a long period containing a change, this change is probably reflected in the corresponding $S_r$ values as well. In case change scenarios are used, these scenarios can include a change in precipitation and/or discharge and (additionally) a (relative) change in transpiration. With respect to the land-use change analysis, this can include a hypothesised change in

transpiration as well. In addition to this, it is likely that different vegetation types adjust to different return periods (Wang-Erlandsson et al., 2016), so a change in land-use can in that way have an effect on the estimated $S_r$-values.

In our opinion it may be worthwhile to include the estimation of (changed) $S_r$ values in these types of analyses, as they could give more information about how the hydrology of a catchment changes under the studied climate or land-use scenarios.

We have elaborated this aspect further in Section 2.3 of the revised manuscript.

**Detailed comments:**

*P1L17: Check the causal order of the mentioned processes "Retreating..."*

We have modified the sentence as follows: 'Increasing temperatures and precipitation, shifts in precipitation from snow to rainfall and retreating seasonal snow cover are a few examples of alterations of the boreal hydrological cycle (Bring et al., 2016).'.

*P2L2: add references*

We have added supporting references, namely:

- Laudon et al. 2011. Consequences of More Intensive Forestry for the Sustainable Management of Forest Soils and Waters. Forests 2, 243-260.

- Nieminen et al. 2017. Impacts of forest harvesting on nutrient, sediment and dissolved organic carbon exports from drained peatlands: A literature review, synthesis and suggestions for the future. Forest ecology and management 392, 13-20.

*P2L7: "but so far none have studied changes in transpiration (patterns) at the catchment scale in boreal regions." Please check (Jaramillo et al., 2018; van der Velde et al., 2013).*

We have changed the sentence as follows: 'Especially under these changing conditions, a proper hydrological understanding of boreal catchments is needed (Waddington et al, 2014; Laudon et al., 2017) to understand the sensitivity and resilience of catchments (Tetzlaff et al., 2013), but also to assess the effect of possible measures. Many studies have been conducted to explore hydrological changes resulting from land use activities (Ide et al., 2013; Mannerkoski et al., 2005, Nieminen et al., 2017), and some already studied changes in transpiration (patterns) at the catchment scale in boreal regions (e.g.van der Velde et al., 2013; Jaramillo et al., 2018)'.

*P2L17: "Thus, climate (or the balance between precipitation and transpiration) has a large influence on the developed $S_r$." Doesn't transpiration depend on the root zone storage (and not the other way around)?*

Yes, we agree with you that transpiration is sustained by the root zone storage capacity and in that sense influenced by it. This is reflected in the used method by assuming that the vegetation has developed a root zone storage capacity to sustain the transpiration demand. However, to calculate this required root zone storage capacity, the long term water balance is used to estimate the transpiration demand of the vegetation. This entire approach assumes equilibrium in the catchment and therefore can be seen as working in two directions: if

either the root zone storage capacity or the transpiration demand changes, the other will (probably) change as well. We have made this clearer in the extended description of the method (see also our reply to the comments of anonymous referee #1).

*P4L3ff: To my knowledge there is a significant undercatch of precipitation, especially in winter. It is not clear if the undercatch was corrected for, but if not, then I disagree with the choice of the authors to correct SWE with P.*

We used a spatially interpolated dataset with a resolution of 10 x 10 km2 for the meteorological parameters (precipitation, air temperature) constructed by Finnish meteorological institute (FMI). In this data set the measurement error caused by gauges has been checked and corrected in operative quality control. For snow data ($S_{SWE}$), we used snow line data provided by Finnish Environment Institute and measured by standard methods. Since $S_{SWE}$ was closest available and not always situated within the study catchment, we corrected $S_{SWE}$ with local precipitation. We have added the following sentence to Section 2.2 to clarify this: 'These data have been checked for measurement errors caused by gauges and were corrected in operative quality control.'

*Sect 2.3 climate derived root zone storage capacity. Since the results show how climate variables correlate with S, I recommend to repeat the key equations to show how climate input is used in the method. Then also the choice of a return period of 20yr may become more clear.*

We have included more details about the used method in the revised manuscript (see also our reply to the first reviewer). The choice of a 20 year return period follows from the analysis of Gao et al. (2014), who found that on average it is most likely that vegetation adapts it root system to a drought with a return period of 20 years.

*P5L12: wording "transpiration demands" is unclear to me*

Transpiration demand is used for the long term deficit between precipitation and discharge. The vegetation in the catchment should have transpired this amount of water to close the long term water balance with the given precipitation and discharge. We have made this clearer in the extended description of the method to estimate $S_r$.

*Results / Discussion: report correlation and significance in text. For example in Sect. 3.1*

The significant correlation coefficients are added as titles to the subplots. In the text it is described whether correlations were significant or not and if so, whether they were positive or negative.

*P8L21: check argument: "The presented results show that climate derived root zone capacities are related to vegetation characteristics, climate variables and vegetation cover, which strongly indicates that the $S_r$-method can be used for boreal regions containing seasonal snow cover." Since S is computed from climate data, the relationship is not a verification of the method!*

We agree with you that this statement is not well formulated; the relationship between climate data and variation of $S_r$ values is indeed not a verification of the method. However, as discussed earlier, we think it valuable to incorporate the comparison with some climate variables. In the revised manuscript we have changed the argument into: 'The presented results show that among the compared characteristics the climate derived

root zone storage capacities are strongest related to climate variables, followed by vegetation characteristics and vegetation types.'

*P9L2: unclear from results "This seems to indicate that in case of low transpiration demands the plant's resources between below and above soil elements are more equally divided than for areas with higher transpiration demands."*

In Figure 3a a larger range of leaf cover/tree height (above ground biomass) can be visually observed for smaller $S_r$-values (below ground biomass) ($S_r < 115$ mm). For larger $S_r$-values, the range in leaf cover/tree height is smaller. This can indicate that the vegetation uses more resources for below ground biomass in cases of larger $S_r$ values. As the derived $S_r$ values are strongly determined by the transpiration demands, the catchments with large $S_r$ values also have high transpiration demand.

As this finding can only be visually observed and was not supported with numbers in this study we have removed the statement from the discussion.

*P9L6f: unclear argument "However, for pine in mid- and south-boreal regions a negative correlation was observed, which means that the vegetation is able to create a larger storage capacity with fewer or thinner roots." Please calculate the significance of the correlation and possibly use a bootstrap to check the influence of outliers. Please check/report how root biomass was calculated. Also check for other influencing variables.*

Although some negative correlation between pine RBM and $S_r$ can be visually observed for the mid- and south boreal regions, this correlation is not significant. However, when the three boreal regions are considered together, the correlation is significant and negative. Which is an interesting aspect to discuss. We have changed the sentence as follows: 'However, for pine a negative correlation was observed, which means that the vegetation is able to create a larger storage capacity with fewer or thinner roots.'

*P9L12: please provide references for shifting management activities*

We have added the following reference in the revised manuscript: Hasper et al. 2016. Water use by Swedish boreal forests in a changing climate, Functional Ecology 30, 690-699.

There is now specific reference for shifting management activities. However, when forest resources are growing faster due to changing climate also forest management activities shifts.

*P10L5: please provide references*

We have added the following reference to support this statement: Menberu M, Tahvanainen T, Marttila H, Irannezhad M, Ronkanen A-K, Penttinen J, Kløve B. 2016. Water table-dependent hydrological changes following peatland forestry drainage and restoration: Analysis of restoration success. Water Resources Research, 52(5), 3742-3760.

*P10L8: "Peatland drainage for forestry changed this pattern: higher $S_r$ values were observed in areas with larger cover of drained peatlands (Figure 7)." I could not see this effect!?*

As discussed earlier, there can be two reasons for the difference between drained and pristine peatlands. We have changed this sentence as follows:'Catchments where peatland is drained for forestry show another pattern: the correlation with $S_r$ is lower, but especially the threshold seems to be weaker. The variation between the

two groups for the threshold analysis is larger for pristine peatlands than for drained ones (Mann-Whitney U-test, p=0.0008 and p=0.0135 respectively).'

*Sect. 4.4: Explain how the method is applied to a change scenario when data on transpiration is required a priori?*

Data on transpiration is indeed used in the analysis; however, this data is derived from the long term water balance (precipitation, discharge and $S_{SWE}$). When the change scenarios are constructed, the transpiration can again be estimated from the water balance or be assumed to change in a certain way. By subsequently calculating $S_r$ values, the effect on the hydrology of the changing conditions can be further explored. We agree that this is not yet a complete analysis, but we definitely see a potential for further research.

We have added a paragraph to Section 2.3 explaining this aspect in more detail.

*Figure 1: Missing y-axis labels; Add points to the boxplots. Panels of Fig1 are insightful, but hardly touched in text. Add relevant topographic info to the map.*

We have added a north arrow to the map and outliers are present for the boxplots. However, adding topographical info made the figure less clear to read, so we prefer not to include it. Similarly, adding labels to the y-axes would mean the figure has to reduce in size, which decreases readability, so we prefer to keep the description in the caption only. A brief description of the panels is included in Section 2.1.

*Fig 2: use white text in dark boxes*

We have changed this in the revised version of the manuscript

*Fig 6c, Fig 8: Julian date for snow off in Fig.6 and Julian Date for max SWE in Fig 8. Please be consistent.*

We have changed all these to 'day of the year', as suggested by anonymous referee # 1.

*Fig 7c,d: Peatland area per catchment? Why does the number of points change?*

Figure 7c illustrates the percentage of the catchment covered with drained peatlands and Figure 7d shows the same for pristine peatlands. The number of points change since some catchments do not have pristine peatland areas and vice versa. We added a note to the caption of the figure to make this clear.

*Fig 8: show correlation as text in one of the diagonals*

We preferred to mention the correlation values with the scatter plots to prevent this figure from overflowing with information. Further, we have moved this figure to the supplement as it has a lot of overlapping info with the principal component analysis presented in Figure 3 in the revised manuscript.

*Fig. 9: What is the ordering in y-axis? Coloring: black lines are hardly seen on dark blue background. Why is PET always the same?*

The ordering on the y-axis is by increasing estimated $S_r$ value and the figure does not show the amount of potential evaporation, but the period in which $E_p$ is occurring/measured. We have clarified this in the figure caption and we have changed the colour of the lines to white.

**3 Reply to review of anonymous referee #2**

Dear referee,

Thank you for reviewing our manuscript, your positive evaluation of the relevance and presentation of the results and the relevant questions you posed. The more detailed comments you have given made us realise that the aim we have in mind with the paper needs to be discussed better throughout the manuscript and that especially some elements of the used method need more attention in a revised version of the manuscript. Below we have replied in more detail to all your comments.

*The study is on an important and interesting topic, as little is known about potential changes in catchment storage properties under climatic and land use change. Potential to improve our hydrological predictions under climatic and land use change is limited by lack of information and understanding, so studies in this area can be expected to be in demand by HESS audience. Another strength of this study is the dataset, which is very well described and referenced. The article is nicely structured, and the results are presented well (though I would prefer to see more numerical information to back up some claims).*

*My main concern about this study is about how much of the results originate from self-prediction given the high correlation between source and comparison data. $S_r$ is derived based on climatic records, and then $S_r$ is compared with climatic and vegetation properties which are known to be related with the source data $S_r$ was derived from. The authors do acknowledge the relationship (e.g. p 5 l 24-25, p 6, l 2-3, p 9 l 16-17), but they still interpret results in a way where (higher) correlation implies control over $S_r$, which I think is questionable. The results might reflect just the closer correlation with the source climatic data for $S_r$, and not causal relationship with the soil storage properties. For example, it remains unclear to what extent the relationship between $S_r$ and vegetation properties/land use are just a consequence of both being related to the climate. In this case the change in vegetation would not influence $S_r$ as it can be expected from the results, if the vegetation is the only thing changing. This would particularly apply to cases where the results are somewhat counterintuitive (e.g. p6 l26-27 and Fig 4a, or p7 l23-25 and Fig 7b where decrease in forested area is associated with increase in $S_r$).*

We agree with you that the variables used in the analysis are subject to internal correlations. As discussed in the reply to the review of Maik Renner, the exact variables used for the calculation of $S_r$ are not the same as those used in the remainder of the analysis.

With respect to the influence of vegetation on $S_r$, this can be reflected by using a different drought return period (Wang-Erlandsson et al., 2016): different vegetation types probably have different survival strategies and therefore are likely to adjust to a different drought return period. However, for this study we only used a 20-year return period as the majority of the land cover consists of forest. So, if only the vegetation would change, a different return period can be used, which would influence the derived $S_r$. However, more testing would be required to see if changes can be assessed by using different return periods and how (quickly) new equilibriums would be established.

In addition to this, the catchments with a lower forest cover are generally the ones with a higher agricultural cover and a milder climate. These catchments are likely to have higher transpiration demands, leading to higher $S_r$ values, than colder forested catchments.

Having said this, the term 'control' might be misleading, especially as our main aim with the study is to compare the calculated $S_r$-values with a set of catchment characteristics to explore possible relations and better understand the derived $S_r$-values and how the different climate variables influence the calculation. Your comments and those of the other reviewers made us realise that we did not discuss this aim consistently throughout the paper. Among others we have changed the title and the last paragraph of the introduction to make this clearer.

*In this light, I think it would be more informative and would give more confidence in the results to apply some method which can account for a number of potential "controls" and assess their importance against each other, for example PCA or multimodel inference (e.g. Saft et al, 2016).*

Thank you for this suggestion; we think as well that the paper would benefit from such further elaboration. Both multimodel inference (as used by Saft et al., 2016) and PCA are useful methods for this. In line with our methodology, we have worked this out further using the PCA. However, we do believe that the multimodel inference is very interesting when evaluating the possibilities to use climate derived $S_r$-values to assess the effects of change.

The results of the PCA are presented in Figure 3 of the revised manuscript, containing the different variables we compared with $S_r$ in the manuscript. We have added section 2.4.4 and 3.1 describing the set-up, results and consequences of the PCA.

*I am also a bit puzzled about the gap between snowmelt and onset of PET, as both are governed by exactly the same increasing energy flux (temperature/sunshine). I would assume that this gap should be very closely related to the maximum SWE ($\sim$ more snow takes longer to melt). Anyway, it would be interesting to calculate this gap (using some threshold for snowmelt) and include it directly as yet another factor along with the other characteristics used. I wonder why it was treated separately.*

$E_p$ and snowmelt are indeed governed by the same increasing energy flux. However, the gap exists because of the measurement methods: $E_p$ is based on pan-evaporation and can thus only be measured if temperatures are above zero. Therefore, $E_p$ can already be slightly above zero before the pan measurements start.

With respect to treating both variables separately, this was done because in the $S_r$ calculations one determines the water demand, the other the water supply. The balance between these variables mainly determines the calculated $S_r$. So, although they are governed by the same energy flux, they have a different effect on the calculation. For the aim of the paper, we think this influence is interesting to investigate and explore.

Having more insight into the combined effect of the onset of $E_p$ and snowmelt can help to assess what can happen in case of a changing climate. For the gap between these two to change, the most important variable will indeed probably be the maximum $S_{SWE}$.

The gap between snow-off and the start of $E_p$ (measurements) is incorporated in the PCA presented in Figure 3 of the revised manuscript. It can be seen this property is strongly positively correlated with the day and amount of maximum $S_{SWE}$. This is further discussed in Section 3.6.

*On a different note, it would be good to see more numerical information (i.e. Spearman's rho, and associate p value) associated with positive/negative correlations described in the text. It is difficult to extract relevant*

*information from figure 8, especially since it is not numeric. Fig 8 also does not include correlation results for sub-regions which are mentioned in the text, and I could not find these results anywhere else.*

As discussed in the replies to the two other referees as well, we have included more numerical information about the correlations in the titles of the scatter plots. We preferred not to include them in Figure 8, to prevent it from overflowing with information.

**Importance and implications:**

*What is the use of the derived $S_r$ and discovered relationships with other characteristics? And in the context of climate change, would not it be easier to derive new $S_r$ following the original method accounting for climate change in the source data instead of looking at the correlations?*

As discussed before, our main of the paper is to explore the relations between the climate derived $S_r$ and a set of catchment variables. We should keep in mind that $S_r$ is a conceptual parameter, originally used as input for hydrological models. However, it is very interesting to know if and how the calculated $S_r$ is related to other variables and if it can be wider applicable. Knowing more about the relations between $S_r$ and catchment variables can help us to better understand the influences on the $S_r$ calculations and therefore how we can, possibly, use it to assess the behaviour of catchment under changing conditions.

Thus, by looking at the relation between $S_r$ and catchment variables, more confidence can be obtained in the (physical) meaning of $S_r$. The found relations are not directly meant to assist to assess change. To assess change indeed recalculating $S_r$ based on new climate predictions would be the most logical approach.

We have made the division between better understanding $S_r$ and using $S_r$ for assessing effects of change clearer in the revised version of the manuscript, among others in the last paragraph of the introduction and the first paragraph of the discussion.

**Specific comments:**

*p2 l 8-10 – and vegetation WUE / transpiring properties*

The partitioning is indeed influenced by both water availability (in the root zone storage) and water use efficiency of the vegetation. However, with the vegetation surviving in a certain catchment, it must have had sufficient water supplies and at the same time it is not logical that it would have invested more carbon in creating storage capacity for water than it needed to survive. So, by deriving the transpiration demand from the water balance, the long term water use is estimated, making the water efficiency of the vegetation less relevant for the calculation. The balance between transpiration and runoff is of course influenced by the water use efficiency of plants.

We have acknowledged this aspect in the introduction and discussion of the revised manuscript, together with a more extensive description of the calculation method for $S_r$ and the assumptions involved.

*p2 l 17 If you talk about climate, do you mean balance between evaporation and precipitation? Transpiration is not purely climatic.*

Actually we are talking about the difference between the long term average supply and demand of water to and from the active storage of the soil. These are mainly climatically driven (ie. via precipitation and potential evaporation). As processes like interception and snow melt are important as well, we decided to use the terms

infiltration and transpiration (demand). Where infiltration is the total precipitation (rainfall and snow) minus interception evaporation and the transpiration (demand) is derived from the water balance ($\overline{T} = \overline{P} - \overline{E_i} - \overline{Q}$). We have clarified these terms in the extended description of the climate derived root zone storage capacity (Section 2.3) in the revised manuscript.

*Section 2.1 – Just checking, is there any permafrost in northern catchments, and if so, can there be any impact (e.g. thawing permafrost → higher storage)?*

Thanks for asking this clarification, but in these sites there is no permafrost. If there would be, changes in permafrost should indeed be included in the calculation of $S_r$, just like snow storage is.

*p4 l 1 – how it was calculated?*

Data of all three biomass variables (root biomass, tree height and leaf cover) are based on field data from national forest inventories, satellite images, digital map data and other georeferenced data sets. More information can be found in Mäkisara et al. 2016 (http://jukuri.luke.fi/handle/10024/532147). We have added this information to Section 2.2 of the revised manuscript.

*Formula 2 – why in the middle line Pi = 1? What does 1 mean?*

Thank you very much pointing this out: it should be Pi = 0. We have corrected this in the revised manuscript.

*Section 4 – Can the changes in $S_r$ be related to changes in WUE (e.g. Troch et al 2009)?*

In this study we did not yet incorporate any changes in the catchments, so we suppose you mean the variations in $S_r$ between the different catchments. The calculation of $S_r$ is based on a daily simulation of soil moisture deficit and an extreme value distribution. The input into the simulation is effective precipitation and a transpiration demand which is estimated from the water balance. As the transpiration demand is the water that should have transpired to close the water balance, different water use efficiency probably will not really influence the derived $S_r$ values. However, water use efficiency could influence the amount of biomass production (root biomass, leaf cover, tree height). Thank you for this suggestion; we have briefly discussed this in Section 4.4 of the revised manuscript.

*p9 l 8-9 – Is it just direct numerical effect of having higher runoff from drained peatlands?*

One of the assumptions underlying the climate derived root zone storage capacity is that a certain type of vegetation needs a specific amount of water to survive; independent of the climate they are located. If they are located in a drier or more seasonal climate, they will need a larger storage capacity to supply the required amount of water. This does neglect the fact that vegetation may have higher water use efficiency (Troch et al., 2009). As discussed earlier, this does not influence the calculation, as the used transpiration demand is derived from the water balance. By deriving the transpiration demand from the water balance, the runoff is already accounted for. However, differences in water use efficiency could help to explain the pattern found for the pine root biomass. We have discussed this in Section 4.2 of the revised manuscript.

*p9 l 11 – suggest changing 'many affects to' to 'many effects on'*

We have changed this in the revised version of the manuscript.

*p9 l 15-16 – I still struggle with the idea of how $S_r$ calculated with pan evaporation would change if only vegetation properties change (as 'or' implies independency) – see my general comment in the beginning. In any case, the argument is based on the assumption of trading space for time (Wagener et al, 2007, Singh et al, 2011), and this and associated assumptions can be acknowledged better (possibly also in introduction).*

$S_r$ is based on a daily simulation of soil moisture deficit and a drought return period the vegetation adapts to. Different vegetation types are likely to adjust to different return periods, as they have different survival strategies. In addition to that, $E_p$ data is only used to add seasonality to the long term averaged transpiration demand, which is derived from the water balance (precipitation and runoff). So, a change in vegetation probably works in two ways: a different drought return period can be applicable and of course the balance between precipitation and runoff can change. In either way, a new equilibrium needs to be established.

In addition, the principle of trading space for time can be used as well, especially as we see shifting conditions in the study areas. However, to go into that direction in this paper would make it lose focus, but we will make the change in vegetation properties we had in mind clearer. Although we have discussed the change part in the paper, and we definitely think a thorough understanding of a climate derived $S_r$ can help to assess hydrological change, our results do not discuss any of these elements.

Having said this, in the revised manuscript we have made clear that the aim of the paper is to better understand the climate derived $S_r$ and its relation with certain catchment properties and we have mentioned the change as an outlook and an, in our opinion, important possible applicability.

**4 Marked-up manuscript**

The following pages contain both the manuscript and the supplement with all changes marked.

[revised manuscript text omitted]

**Figure 7.** [This figure is updated with correct data and was Figure 3 in the original manuscript] Calculated root zone storage capacity versus average leaf cover (top) and tree height (bottom) of four years. Larger circles indicate higher percentage of vegetation type for a&e) forest, b&f) pristine peatlands, c&g) agriculture; d&h) are colour coded by boreal region. $S_r$ has statistically significant Spearman's correlation with leaf cover ($r = 0.33$) and tree height ($r = 0.32$). Different boreal regions did not resulted in statistically significant correlations when considered individually.

[Figure]

**Figure 8.** [This was Figure 7 in the original manuscript] Root zone storage capacities ($\cancel{S_{r,20}} \underline{S_r}$, mm) and proportion of a) agricultural areas (%), b) forest cover (%), c) drained peatlands (%) and d) undrained peatlands (%) in the catchment at different ecoregions (S is south boreal, M is mid-boreal and N is north boreal). The titles of the subplots show the Spearman's correlation coefficients (significant correlation for p<0.05). Note that not all vegetation types are present in each catchment, thus the number of catchments per subplot may differ.

The correlation matrix (originally Figure 8) was moved to the supplementary material

**Figure 8.**

[Figure]

**Figure 9.** [this figure has been changed: black lines are replaced by white lines and labels on y-axes have been adjusted] Snow cover is presented by the colour plot (red: $S_{SWE}$ > 15 mm, blue: $S_{SWE}$ = 0)Occurrence of potential evaporation ($E_p$ > 0) is presented by white lines; note that the actual amount of $E_p$ is not presented. Presented data are long term daily averages. Catchments are ordered by increasing $S_r$-values.

**Supplement belonging to " Understanding variability in root zone storage capacity in boreal regions"**

by Tanja de Boer-Euser, Leo-Juhani Meriö, Hannu Marttila

**1  Background on study catchments**

[revised manuscript text omitted]

[This figure was moved from the manuscript to the supplement]

[Figure]

**Figure 1:** Correlation matrix for calculated root zone storage capacity (20 year return period), calculated transpiration demands (used in the $S_r$ calculation) and catchment characteristics. The sizes of the boxes indicate the p-values; the asterisks indicates a significant correlation (p<0.05).

---

## Referee Report (RR1)

Overall comments:

The authors have come back with an improved manuscript that addresses many of the comments and concerns raised in the initial assessments. I am recommending publication but still believe that some changes are necessary. I am not sure that the authors have established "control" (i.e. page 1 line 5) versus "correlation", especially given the intrinsic relationships and correlations between the variables used in calculating $S_{r,20}$ and the assessment. I do appreciate the analysis conducted and do not dismiss the results. I believe there may be merit in the hypothesis, which warrants future studies in different locations. I don't believe it is necessary to come to a strong conclusion in this paper. What I would like to see is a more frank discussion of the caveats and limits to what has been presented, but with a view towards the possibilities associated with extensions of this work, which may indeed prove useful.

Specific Comments:

Figure 7: There appear to still be inconsistencies in the data presented. In the mid-boreal region there is a catchment with almost 60% drained peatland and $S_{r,20}$ of around 40 mm. However, while there are some points representing pristine peatlands with a similar $S_{r,20}$, there are no corresponding lower percentage points for forest cover and agricultural areas. I am speculating that that there are some zero points or points very close to zero that are not shown. Where are the corresponding fractions of forest and pristine peatlands for these points? There apper to be other such examples. The removal of zero or near-zero points (some or all?) might be biasing the interpretation of results. Am I reading the plots incorrectly?

Page 6 line 20-21: I suspect that the correlation between $S_{r,20}$ and leaf cover or tree length for the entire dataset is better than for any of the three regions. It appears that the forest structure follows a rough latitudinal gradient (the authors have noted climate effects in both directions in their previous response). Separating the data into regions defined by latitude makes the relationships harder to see because some of the main drivers associated with latitude are excluded. The figure does not need to be changed but this point could be added.

Page 6 lines 25-27: Figure 4 may be showing the 'preference' of spruce, pine and deciduous trees. Pines often locate on sandy or rapidly draining soil, which has a small $S_r$ because the difference between field capacity and the wilting point is small. Pines do not grow well in wet soils so the largest root biomass is likely found where they grow well. Spruce trees do not grow as well in sandy dry areas so the largest root biomass is found in the areas with more moderate drainage. In the north where there are thick peat soils, these soils developed because of the persistent poor drainage and high water table (and slow decomposition). If the peat soils did not dry out during the 20-year drought periods, they will have small estimated $S_{r,20}$ values and since trees do not grow deep roots in waterlogged soils this creates an association with trees in low $S_{r,20}$ regions having a low root mass, except for pines which are in areas with a small $S_{r,20}$ for different reasons described above. I see this sort of discussion as helpful in understanding the multi-facetted relationship between the climate, soil, hydrology and vegetation, rather than implying undue control by vegetation.

Page 7 line 15-17: The words "leads to" implies causation which I am not sure exists. I suggest changing "leads to" to "is associated with" which is softer and less open to criticism. Some discussion about causal relationships and correlation is warranted in the paper overall. Tree growth is affected by seasonal and annual temperatures as is evaporation, rain/snow partitioning and the snow off date.

Page 7 line 23-24: I am not asking for these to be answered directly but the authors might consider them and add to the discussion only if relevant to this region and the results. What role do natural drainage (or lack hereof) caused by soil depth, soil texture or topography and the effect of temperature on evapotranspiration play in the $S_{r,20}$ values? Is $S_{r,20}$ greater in the south because evaporative demand is higher and forest cover smaller because the forests were cleared and/or planted on drained peatlands? Are forested peatlands classified as forests or peatlands? I have read that many of the drained peatlands were planted as forests. In which category are these included?

Page 8 line 11-19: I present this as an alternative to the threshold interpretation. For catchments with more forest, the snowpack is sheltered by the forest and the snow melts later but the air temperature is warmed by the dark canopy albedo. So the snow in the northern forest probably melts at warmer (weather station) air temperatures than in the south. I suspect that the evaporation pans are generally placed at weather stations located in the open, not under the shelter of forests, so they will experience the warmer air (relative to the cold sub-canopy air) and Ep will start while the snow is still on the ground under the canopy. Where there is less forest the more exposed snow melts faster and more in-line with increases in air temperature and solar radiation and is gone before Ep becomes significant. I am not certain whether the differences in forest cover are enough to be the main driver of this apparent relationship with Ep and $S_{r,20}$ but if it contributes it supports softening the conclusions.

Page 8 line 21: I believe correlation has been shown and the results are intriguing but I don't believe causation has been successfully argued.

Page 9 lines 1 and 2: The sample size of catchments with small agricultural leaf cover appears too small to make inferences about the effect of varying agricultural leaf cover. However, I am wondering where the zero values are? Are there no catchments with zero or near-zero agricultural cover, zero or near zero pristine peatlands?

Page 9 lines 6-9: While the authors do acknowledge that the density of pine trees may be too low in these catchments to have much influence on transpiration and storage, I still feel that the preceding statement goes too far. The authors do not present any information about the percentage or proportion of leaf cover or tree cover that is represented by pine, spruce and deciduous species. Are the RBM values calculated on a catchment basis, such that the sum of root biomass for each species in each catchment is divided by the entire area of each catchment, or is the root biomass merely the average for each species within the areas that contained that species (i.e. within sub-areas of each catchment, the fractions of which we do not know)? If these RBM values are averaged over the area of the catchment, then I would interpret Figure 4a as showing that pine do not grow well in catchments that have a very high or very low $S_{r,20}$. A very low $S_{r,20}$ might indicate a lack of sufficient soil, or perpetually water-logged conditions, such that all trees grow poorly. At slightly greater but still small $S_{r,20}$ values, we may see the

sandy areas favoured by pines represented; they grow well but need a high root biomass to access enough water in the rapidly draining sandy soil. As $S_{r,20}$ continues to increase, we likely encounter conditions in which the soil is wet enough of the time that spruce and deciduous species outcompete pine.

Page 10 line 4-7: Yes, the boreal ecosystem has been referred to as a "green desert". (Hall, 1999, https://doi.org/10.1029/1999JD901026; Betts et al. 2001, 2001 JD900047). There is ample water on the surface but either because of nutrient limitations or adaptation to cool environments, the vegetation is less productive and evaporation rates are generally low.

Page 10 line 16-18: The authors should expand on this idea of a climate-derived $S_{r,20}$ being useful to assess the hydrological effect of future changes in climate and land cover. This point could provide justification for this work. Can the authors discuss what would be required for this to happen? Additional studies need to be conducted to assess the usefulness of a climate-derived $S_{r,20}$ and its applicability in different locations. Then, this could be evaluated in models of sufficient complexity, and if the patterns are similar, in climate change scenarios. At this stage, stating that this method is useful is conjecture, but pointing to work that would serve to test this would set this paper in a better context.

Corrections:

Page 1 line 3: Change "enables to account" to "enables one to account".

Page 1 line 11: Change "besides from" to "apart from".

Page 1 line 20: change "alter magnitude" to "alter the magnitude".

Page 1 line 23: Change "in near future" to "in the near future".

Page 1 line 24: Awkward sentence. Change "The occurring land use changes consist of" to "These land use changes consist of".

Page 2 line 2: Change "as source for biomass" to "as a source of biomass".

Page 2 line 5: I am not sure what the authors mean by "measures".

Page 2 line 21: Change "boreal catch" to "boreal catchments".

Page 3 line 2: Change "Characteristics study catchments" to "Characteristics of study catchments".

Page 3: There are a number of very short paragraphs, some of which could be combined.

Page 3 line 5: Change "belong to National network" to "belong to a national network".

Page 3 line 13 and 14: Change "sites" to "catchments". Unless a specific field measurement location is referred to, the use of 'catchments' is preferred over 'sites' because 'sites' suggests specific locations in space, and I believe all of the data are presented at the catchment scale.

Page 3 line 22: Change "additional data were used about leaf cover" to "additional data were used, including leaf cover".

Page 4 line 1: Change "data was available" to "data were available".

Page 4 line 2: Change "from Finnish" to "from the Finnish".

Page 4 line 6: Change "data was adjusted" to "data were adjusted".

Page 5 line 6: Change "sevaral" to "several".

Page 10 line 11: Change "affects to" to "effects on".

Figure 6a: The use of the term Julian day is not correct. It should be day of year. The authors indicated that this would be corrected but have not done so.

---

## Author Response (AR2)

**Author's response for review process of "Understanding variability in root zone storage capacity in boreal regions"**

by Tanja de Boer-Euser, Leo-Juhani Meriö, Hannu Marttila

Dear editor,

Please find attached the second revised version of our manuscript. The additional comments of the reviewers have been very useful for this revised version. This document first contains a point-to-point reply to all their comments. The point-to-point reply is followed by a marked-up version of the revised manuscript and supplement.

The most important change in the manuscript is the location of the presentation of the PCA results. These are now located after the presentation of the comparison of $S_r$ with vegetation types. Further, textual clarifications have been made, especially in the discussion section.

We would like to submit this second revised version for the next step in the review process.

On behalf of all authors,

Kind regards,
Tanja de Boer-Euser

**1 Reply to review of Maik Renner**

Dear Maik Renner,

Thank you for the evaluation of our revised manuscript and the remaining constructive comments. We have replied to your comments below and incorporated them in the latest version of the manuscript.

*The authors did a thorough revision of their manuscript with improvements in many aspects. The authors also responded to all concerns raised by me and the other two reviewers.*

**1.1 Overall comments:**

*There is, however, one major point which must be addressed in a another revision of the manuscript. It is about the order of the figures and the results. I cannot understand why Fig 1 showing the derived Sr values is mentioned in the methods sections while this clearly a result of this study. Furthermore, the first section in the results is called "dependencies" which is quite unspecific. I would suggest to order Figures and results in a more classic way. For example starting with the scheme of the method as first figure and a map of the region as second figure.*

*The results should start with the maps of the derived Sr values (note the spatial coherence) and the strong differences when looking at the different boreal regions. After this the correlations to climate, vegetation, etc can be shown. The PCA then nicely summarizes the link of all the catchment characteristics to the new catchment storage estimate.*

*Such a reorganization is feasible and therefore I recommend a minor revision. It would clearly improve the presentation of the results.*

Thank you for this suggestion, we agree that the results are indeed more logically presented if the PCA is moved to after the comparisons of $S_r$ with climate variables, vegetation characteristics and vegetation types. So, we have incorporated this change.

However, regarding Figures 1 and 2, we think it is more logical to first present the study area, followed by the details of the method to derive $S_r$ from climate data. However, to keep presentation of the study area and presentation of the results separated, we have removed the derived $S_r$ from Figure 1 and replaced it with forest cover.

**1.2 Minor comments:**

*title sections 2.4.4 and 3.1 Please use a more specific title then dependencies*

We have changed the titles to 'Correlations among catchment characteristics'

*PCA methods and results: It should be noted how the PCA was set up, I believe across all catchments. Also report the explained variance of the first two PCA's. This is important to see how relevant the specific features are.*

Yes, the PCA was set up across all catchments, we have clarified this in the text, together with mentioning the combined explained variance (54%) of the first two principal components. Further we have added a table

Variability in $S_r$ in boreal regions - author's response 2

to the supplementary material containing the explained variance of the first two principal components and the loadings of all used characteristics on the first two principal components.

*P5L20: The choice of the 20 yr return period is quite ad-hoc. Why should it be 20yrs in a boreal region, why should this be constant? I think this is an assumption and it could be worthwhile to discuss the implications of this assumption in the discussion. For example by how much would results change when a different return period is assumed? Does a return period of 20yrs make sense for a agriculturally dominated catchment? Actually Section 4.3 starts with such a discussion but does not link it with the return period.*

Thank you for this comment. The 20 year return period is indeed an average we selected for this study, following the results presented by Gao et al. (2014) and Wang-Erlandsson et al. (2016). For catchments with a large agricultural cover, a smaller return period would be more realistic. However, only 3 of the 64 catchments have more than 50% agricultural cover and 8 of the 64 have more than 25% agricultural cover. Therefore, we think using different return periods in this study will mainly increase the amount of variables, without changing the results substantially.

Having said this, the selected return period has an effect on $S_r$-values derived for the individual catchments, so should be discussed in the discussion of the manuscript. We have done this is Section 4.3.

*Fig 4: maps of Sr. The background colors of the 3 regions is visually more pronounced than the Sr values which are shown by the size of the labels. Maybe also use color for the Sr values and only use dotted lines to disentangle the different regions.*

Thank you for this suggestion, we have changed the figure accordingly.

*FIG 7. Legend for point size is missing*

We have added a legend for the point sizes to the figure.

**2  Reply to review of anonymous referee #1**

Dear referee,

Thank you for the evaluation of our revised manuscript. Your additional comments were valuable for the latest version of our manuscript. However, considering the mentioned line numbers, we have the feeling you have reviewed the first version of the manuscript again, after reading our replies to your comments and those of the other referees. We have replied in detail on your comments below; those not already incorporated in the previous version of the manuscript are now incorporated.

**2.1  Overall comments:**

*The authors have come back with an improved manuscript that addresses many of the comments and concerns raised in the initial assessments. I am recommending publication but still believe that some changes are necessary. I am not sure that the authors have established "control" (i.e. page 1 line 5) versus "correlation", especially given the intrinsic relationships and correlations between the variables used in calculating Sr,20 and the assessment. I do appreciate the analysis conducted and do not dismiss the results. I believe there may be merit in the hypothesis, which warrants future studies in different locations. I don't believe it is necessary to come to a strong conclusion in this paper. What I would like to see is a more frank discussion of the caveats and limits to what has been presented, but with a view towards the possibilities associated with extensions of this work, which may indeed prove useful.*

Thank you for your evaluation. In this version we have elaborated some elements in the discussion further.

**2.2  Specific Comments:**

*Figure 7: There appear to still be inconsistencies in the data presented. In the mid-boreal region there is a catchment with almost 60% drained peatland and Sr,20 of around 40 mm. However, while there are some points representing pristine peatlands with a similar Sr,20, there are no corresponding lower percentage points for forest cover and agricultural areas. I am speculating that that there are some zero points or points very close to zero that are not shown. Where are the corresponding fractions of forest and pristine peatlands for these points? There apper to be other such examples. The removal of zero or near-zero points (some or all?) might be biasing the interpretation of results. Am I reading the plots incorrectly?*

Both in Figures 6 and 7 (numbering according to latest version of the manuscript) all catchments are now included in all subplots, so also if a certain vegetation type is not present in the specific catchment.

*Page 6 line 20-21: I suspect that the correlation between Sr,20 and leaf cover or tree length for the entire dataset is better than for any of the three regions. It appears that the forest structure follows a rough latitudinal gradient (the authors have noted climate effects in both directions in their previous response). Separating the data into regions defined by latitude makes the relationships harder to see because some of the main drivers associated with latitude are excluded. The figure does not need to be changed but this point could be added.*

The correlation coefficients for the different comparisons are now mentioned with the figures. A stronger correlation is indeed present when all catchments are considered; we have discussed this in Section 4.1.

*Page 6 lines 25-27: Figure 4 may be showing the 'preference' of spruce, pine and deciduous trees. Pines often locate on sandy or rapidly draining soil, which has a small Sr because the difference between field capacity and the wilting point is small. Pines do not grow well in wet soils so the largest root biomass is likely found where they grow well. Spruce trees do not grow as well in sandy dry areas so the largest root biomass is found in the areas with more moderate drainage. In the north where there are thick peat soils, these soils developed because of the persistent poor drainage and high water table (and slow decomposition). If the peat soils did not dry out during the 20-year drought periods, they will have small estimated Sr,20 values and since trees do not grow deep roots in waterlogged soils this creates an association with trees in low Sr,20 regions having a low root mass, except for pines which are in areas with a small Sr,20 for different reasons described above. I see this sort of discussion as helpful in understanding the multi-facetted relationship between the climate, soil, hydrology and vegetation, rather than implying undue control by vegetation.*

Thank you for this analysis, we have discussed this multi-facetted relationship further in sections 4.2 and 4.3 of the revised manuscript.

*Page 7 line 15-17: The words "leads to" implies causation which I am not sure exists. I suggest changing "leads to" to "is associated with" which is softer and less open to criticism. Some discussion about causal relationships and correlation is warranted in the paper overall. Tree growth is affected by seasonal and annual temperatures as is evaporation, rain/snow partitioning and the snow off date.*

The mentioned sentence is no longer present in the revised manuscript.

*Page 7 line 23-24: I am not asking for these to be answered directly but the authors might consider them and add to the discussion only if relevant to this region and the results. What role do natural drainage (or lack hereof) caused by soil depth, soil texture or topography and the effect of temperature on evapotranspiration play in the Sr,20 values? Is Sr,20 greater in the south because evaporative demand is higher and forest cover smaller because the forests were cleared and/or planted on drained peatlands? Are forested peatlands classified as forests or peatlands? I have read that many of the drained peatlands were planted as forests. In which category are these included?*

As $S_r$ is derived from climate data, the effect of temperature and evaporation is large. As argued during the previous revision, the used method assumes equilibrium in the catchments. Apparently, the vegetation was able to survive given the occurring precipitation and evaporation, and thus runoff generation in the catchment. To do this, it created a sufficient buffer with its root system. Ground water table levels, soil depth, soil texture and topography likely determine how this root system was developed, but too a much smaller extend its size (ie., its buffer capacity). In our study we only took into account the estimated buffer capacity of the root system as a catchment representative characteristic. It was beyond the scope of the study to consider its internal structure.

Evaporation is higher in the south boreal region because of higher potential evaporation, but forest cover is also higher in the south boreal areas. Our catchments did not contained any large forest cutting areas. In

Finland the majority of peatlands has been drained for forestry purposes. In our study, all forests are clasified as 'Forests'; the drained peatlands in Figure 8 can contain forest, but can also contain other vegetation types. We have clarified this in Sections 2.4.2 and 2.4.3.

*Page 8 line 11-19: I present this as an alternative to the threshold interpretation. For catchments with more forest, the snowpack is sheltered by the forest and the snow melts later but the air temperature is warmed by the dark canopy albedo. So the snow in the northern forest probably melts at warmer (weather station) air temperatures than in the south. I suspect that the evaporation pans are generally placed at weather stations located in the open, not under the shelter of forests, so they will experience the warmer air (relative to the cold sub-canopy air) and Ep will start while the snow is still on the ground under the canopy. Where there is less forest the more exposed snow melts faster and more in-line with increases in air temperature and solar radiation and is gone before Ep becomes significant. I am not certain whether the differences in forest cover are enough to be the main driver of this apparent relationship with Ep and Sr,20 but if it contributes it supports softening the conclusions.*

We agree with hypothesis and it should be valid when compared with catchments within the same climate region (eg. catchments within the southern boreal). Furthermore, to test this hypothesis the comparison should include catchments with full forest cover and catchments with heavy forestry operations (clear cutting). In these circumstances canopy albedo differences and accumulation of snow start to influence on snow melting conditions and further probably to $E_p$ conditions. Nevertheless, in our catchments we did not have the possibility for this comparison and the main driver in maximum $S_{SWE}$ values and $E_p$ was the difference in climate across the regions. One should observe that climate varies notable in Finland from south to north (from temperate to sub-boreal conditions) which dominates variability between the catchments.

In addition to this, we have discussed the possibility that the threshold is the result of a measurement artifact (Section 4.1) and what would be the consequences of this for using $S_r$ to assess hydrological effects of changing climatic and vegetation conditions (Section 4.4).

*Page 8 line 21: I believe correlation has been shown and the results are intriguing but I don't believe causation has been successfully argued.*

During the first revision of the manuscript we have removed the term 'control' from the manuscript, as we have indeed more investigated correlations and no causal relations. Further, in Section 4.2 we discuss that we assume the vegetation to be in balance with the transpiration demand, but that not necessarily one is causing the other.

*Page 9 lines 1 and 2: The sample size of catchments with small agricultural leaf cover appears too small to make inferences about the effect of varying agricultural leaf cover. However, I am wondering where the zero values are? Are there no catchments with zero or near-zero agricultural cover, zero or near zero pristine peatlands?*

This paragraph is no longer present in the revised version of the manuscript.

*Page 9 lines 6-9: While the authors do acknowledge that the density of pine trees may be too low in these catchments to have much influence on transpiration and storage, I still feel that the preceding statement goes too far. The authors do not present any information about the percentage or proportion of leaf cover or tree*

*cover that is represented by pine, spruce and deciduous species. Are the RBM values calculated on a catchment basis, such that the sum of root biomass for each species in each catchment is divided by the entire area of each catchment, or is the root biomass merely the average for each species within the areas that contained that species (i.e. within sub-areas of each catchment, the fractions of which we do not know)? If these RBM values are averaged over the area of the catchment, then I would interpret Figure 4a as showing that pine do not grow well in catchments that have a very high or very low Sr,20. A very low Sr,20 might indicate a lack of sufficient soil, or perpetually water-logged conditions, such that all trees grow poorly. At slightly greater but still small Sr,20 values, we may see the sandy areas favoured by pines represented; they grow well but need a high root biomass to access enough water in the rapidly draining sandy soil. As Sr,20 continues to increase, we likely encounter conditions in which the soil is wet enough of the time that spruce and deciduous species outcompete pine.*

Thank you for this suggestion and argumentation. The root biomass values are indeed averaged over the entire area of the catchment. The mentioned statement is no longer present in revised manuscript, but we have discussed this aspect at the end of Section 4.2.

*Page 10 line 4-7: Yes, the boreal ecosystem has been referred to as a "green desert". (Hall, 1999, `https://doi.org/10.1029/1999JD901026`; Betts et al. 2001, 2001 JD900047). There is ample water on the surface but either because of nutrient limitations or adaptation to cool environments, the vegetation is less productive and evaporation rates are generally low.*

Thank you for this suggestion. We have mentioned the concept of "green desert" and the suggested references in the beginning of section 4.

*Page 10 line 16-18: The authors should expand on this idea of a climate-derived Sr,20 being useful to assess the hydrological effect of future changes in climate and land cover. This point could provide justification for this work. Can the authors discuss what would be required for this to happen? Additional studies need to be conducted to assess the usefulness of a climate-derived Sr,20 and its applicability in different locations. Then, this could be evaluated in models of sufficient complexity, and if the patterns are similar, in climate change scenarios. At this stage, stating that this method is useful is conjecture, but pointing to work that would serve to test this would set this paper in a better context.*

Thank you for this suggestion for a better framing of the usefulness of a climate derived $S_r$ to assess the hydrological effects of changing climatic and vegetation conditions. We have expanded the last paragraph of section 4.4.

**2.3 Corrections:**

*Page 1 line 3: Change "enables to account" to "enables one to account".*
We have changed the sentence accordingly.

*Page 1 line 11: Change "besides from" to "apart from".*
We have changed the sentence accordingly.

*Page 1 line 20: change "alter magnitude" to "alter the magnitude".*

We have changed the sentence accordingly.

*Page 1 line 23: Change "in near future" to "in the near future".*
We have changed the sentence accordingly.

*Page 1 line 24: Awkward sentence. Change "The occurring land use changes consist of" to "These land use changes consist of".*
We have changed the sentence accordingly.

*Page 2 line 2: Change "as source for biomass" to "as a source of biomass".*
We have changed the sentence accordingly.

*Page 2 line 5: I am not sure what the authors mean by "measures".*
We have replaced 'measures' with 'land use activities'.

*Page 2 line 21: Change "boreal catch" to "boreal catchments".*
We have changed the sentence accordingly.

*Page 3 line 2: Change "Characteristics study catchments" to "Characteristics of study catchments".*
We have changed the header accordingly.

*Page 3: There are a number of very short paragraphs, some of which could be combined.*
We have combined the last two paragraphs. The other ones we prefer to keep separate, as they treat different aspects of the study catchments.

*Page 3 line 5: Change "belong to National network" to "belong to a national network".*
We have changed the sentence accordingly.

*Page 3 line 13 and 14: Change "sites" to "catchments". Unless a specific field measurement location is referred to, the use of 'catchments' is preferred over 'sites' because 'sites' suggests specific locations in space, and I believe all of the data are presented at the catchment scale.*
We agree with you that 'catchments' is preferred over 'sites' and have changed this throughout the manuscript.

*Page 3 line 22: Change "additional data were used about leaf cover" to "additional data were used, including leaf cover".*
We have changed the sentence accordingly.

*Page 4 line 1: Change "data was available" to "data were available".*
We have changed the sentence accordingly.

*Page 4 line 2: Change "from Finnish" to "from the Finnish".*
We have changed the sentence accordingly.

*Page 4 line 6: Change "data was adjusted" to "data were adjusted".*

We have changed the sentence accordingly.

*Page 5 line 6: Change "sevaral" to "several".*

In the revised version of the manuscript this sentence is changed and the word is no longer there.

*Page 10 line 11: Change "affects to" to "effects on".*

We have changed the sentence accordingly.

*Figure 6a: The use of the term Julian day is not correct. It should be day of year. The authors indicated that this would be corrected but have not done so.*

We have corrected this in the figures and text.

**3 Marked-up manuscript**

The following pages contain both the manuscript and the supplement with all changes marked.

[revised manuscript text omitted]

[c5]https://hakku.gtk.fi/en/locations/search
[c6]http://kartta.metla.fi/opendata/valinta.html

Variability in S$_r$ in boreal regions - author's response 14

[revised manuscript text omitted]

Variability in $S_r$ in boreal regions - author's response 26

[revised manuscript text omitted]

Variability in S$_r$ in boreal regions - author's response 44

**2 Background on correlations between catchment characteristics**

**2.1 Principal component analysis**

Table 3 shows the explained variance of the first two principal components, together with the loadings of all used catchment characteristics on these two principal components.

**2.2 Correlation matrix**

Figure 1 shows the correlations between $S_r$ and the various catchment characteristics. From this figure it follows that the strongest positive correlation was found between $S_r$ and the mean annual temperature and the strongest negative correlation was found for $S_r$ and the (timing of) maximum $S_{SWE}$. Further, it can be seen that a strong correlation exits between the different vegetation characteristics and between the different climate variables. In addition, the different land covers (except for drained peatlands) also show a significant correlation with the climate variables.

[Figure]

**Figure 1:** Correlation matrix for calculated root zone storage capacity (20 year return period), calculated transpiration demands (used in the $S_r$ calculation) and catchment characteristics. The sizes of the boxes indicate the p-values; the asterisks indicates a significant correlation (p<0.05).

**Table 3:**  [This table was added]Summary of principal component analysis (PCA). The highest loads for each characteristic are shown in bold.

|  | PC1 | PC2 |
|---|---|---|
| Eigenvalue | 8.20 | 4.68 |
| % Explained | 34 | 20 |
| Cumulative % explained | 34 | 54 |
| Forest | -0.364 | **-0.825** |
| Conifer | -0.516 | **-0.866** |
| Broadleaved | **0.350** | 0.169 |
| Peatland | **-0.727** | 0.253 |
| Agriculture | **0.796** | 0.509 |
| Precipitation | 0.254 | **-0.944** |
| Summer precipitation | 0.021 | **-0.865** |
| max $S_{SWE}$ | **-1.168** | -0.156 |
| $P/E_p$ | **-0.923** | -0.068 |
| Longitude | **-0.771** | -0.643 |
| Latitude | **-1.086** | 0.544 |
| Leaf cover | 0.025 | **-0.974** |
| Pine RBM | 0.227 | **0.563** |
| Spruce RBM | 0.365 | **0.433** |
| Decidious RBM | 0.294 | **0.534** |
| Total RBM | 0.298 | **0.525** |
| Tree height | 0.528 | **-0.961** |
| Drained peat | **-0.198** | -0.029 |
| Pristine peat | **-0.751** | -0.232 |
| Timing max $S_{SWE}$ | **-1.221** | 0.071 |
| snow-off | **-1.206** | 0.237 |
| mean annual temperature | **1.186** | -0.349 |
| Tdemand | **0.857** | -0.495 |
| Gap $S_{SWE}$ $E_p$ | **-1.100** | -0.041 |

---

## Author Response (AR3)

**Author's response for review process of "Understanding variability in root zone storage capacity in boreal regions"**

by Tanja de Boer-Euser, Leo-Juhani Meriö, Hannu Marttila

Dear editor,

Thank you for the quick evaluation of our revised manuscript. With regards to your final point, we would like to emphasize that indeed we have taken this comment from Reviewer #2 into account seriously and incorporated it into the paper.

We believe (given the mentioned line numbers) that this comment refers to the initial version of the paper (uploaded on 30 March 2018), where indeed such discussion on limitations and future direction was not extensively done. In our second version (uploaded on 20 August 2018) we had already corrected this, by extensively revising the discussion, but also in a more integral manner in the tone and approach throughout the paper. The track changes in this revision provide more detail and guidance on how we have processed this exactly. We also added the PCA, which deals with the intrinsic relationship between the different variables compared with $S_r$. For the latest revision (uploaded on 17 November 2018), we have done another check on this issue – as well as on the other comments - and made the subsequent final improvements to the paper.

Specifically, the issues raised by the reviewer can be found at the following places (section, figures and line numbers refer to the marked-up version of the latest manuscript).

**Control vs correlation** throughout the manuscript the term 'control' is replaced by either correlation or variability (see also first paragraph of page 18 of the author's response uploaded on 20 Augustus).

**Relations between catchment variables and $S_r$** to make these relations clearer, we have added the PCA analysis (Sections 2.4.4 and 3.4) and added correlation coefficients for all assessed relations between $S_r$ and catchment characteristics (Figures 4-7).

**Caveats and limitations of presented results** Section 4 now contains a discussion of the limitations of the findings, for example in section 4.2 (p10, l.28-30) and section 4.3 (p.11, l.18-21) where reasons are discussed for the lack of a clear pattern in the comparison between $S_r$ and vegetation

characteristics and types. Section 4.3 (p.11, l.21-29) poses further a set of questions about remaining interactions between catchment variables and derived $S_r$-values that are not yet considered in this study.

**Relevant studies for future work** Section 4.4 deals with an outlook for future use of $S_r$: it suggests repeating the study in other boreal and temperate climates to further investigate the threshold behaviour (p.12, l.25-27) and it discusses the need for further investigating the way (new) equilibriums between climate and vegetation are established in different catchments (p.12, l.29-31).

I hope this clarifies your doubts.

On behalf of all authors,

Kind regards,
Tanja de Boer-Euser